# Linking chemical weathering, evolution of preferential flow paths and transport self-organization in porous media using non-equilibrium thermodynamics

Evgeny Shavelzon<sup>1</sup>, Erwin Zehe<sup>2</sup>, and Yaniv Edery<sup>1</sup>

**Correspondence:** Yaniv Edery(yanivedery@technion.ac.il)

**Abstract.** Chemical weathering of soil and rock is a complex geophysical process during which the reaction and transport processes in the porous medium interact, causing erosion of the medium. This process is ubiquitous in geophysical systems and can be encountered, among others, in formation of karst systems, subsurface carbon sequestration and surface weathering of river beds. A common outcome of chemical weathering is the emergence and intensification of preferential flow paths, where the weathering alters the transport properties of the rock, thus introducing coupling between transport and reaction. While numerous approaches have been undertaken to simulate this complex interaction, still a need exists for a unified framework able to correlate the emergence of preferential flow paths due to reaction-transport interaction with the associated dissipative dynamics. Here we propose such a framework considering the case of subsurface chemical weathering of calcite porous rock undergoing reversible dissolution-precipitation reaction, and apply non-equilibrium thermodynamics to analyze the ensuing reaction-transport interaction in this geophysical scenario. We identify the entropy generation sources, attributed to the dissipative processes inherent to this physical scenario and show a clear correlation between the emergence and intensification of preferential flow paths and the accompanying dissipative dynamics, where the evolution of the emerging paths leads to a decrease in the free-energy dissipation rate due to flow percolation, mixing of chemical constituents and reaction. This indicates that the emergence of preferential flow paths due to chemical weathering in geophysical systems represents an energeticallypreferred state of the system that can be considered a manifestation of the minimum energy dissipation principle. Our analysis implies that, for a given pressure head, a more homogeneous porous matrix will result in less pronounced preferential flow paths, along with lower flow and higher mineralization rates. On the other hand, for a highly heterogeneous matrix dominant preferential flow paths will be obtained, along with higher flow and lower mineralization rates. Considering these aspects for carbon sequestration where acidified brine leads to carbon mineralization, we conclude that, for a given pressure head, an injection into a more heterogeneous matrix will result in a higher injection rate, while a more homogeneous domain will yield a higher mineralization rate, thus exemplifying the resulting trade-off in the injection strategy.

<sup>&</sup>lt;sup>1</sup>Faculty of Civil and Environmental Engineering, Technion - Israeli Institute of Technology, Haifa, Israel

<sup>&</sup>lt;sup>2</sup>Karlsruhe Institute of Technology (KIT), Institute of Water and River Basin Management, Karlsruhe, Germany

#### 1 Introduction

# 1.1 Chemical weathering as a non-equilibrium thermodynamic process

Chemical weathering of soil and rock, ubiquitous in the field of Earth science, is a complex geophysical process during which the reaction and transport processes in the porous medium interact, causing erosion of the medium (Marshall and Fairbridge, 2018). One of the main forms of subsurface chemical weathering involves dissolution-precipitation of the porous medium induced by the invading fluid, thus affecting its transport properties by altering pore sizes and their connectivity, and leading to the formation of conductive channels (Kamolpornwijita et al., 2003, Zhang et al., 2021). This introduces coupling between the chemical reaction and transport, leading to the emergence and intensification of the preferential flow path phenomenon, when the flow and transport are being funneled into distinct flow paths. From a broader perspective, this "chemical erosion" is comparable to the emergence of rill and river networks through mechanical erosion, which similarly establishes and enhances distinct flow paths within the network (Berkowitz and Zehe, 2020).

The emerging preferential flow paths act as the main conductive channels in a porous medium, facilitating solute transport (Beven and Germann, 1982, Stamm et al., 1998, Edery et al., 2016) and intensifying the transversal concentration gradients (Zehe et al., 2021). Stronger transport channelization into preferential flow paths is reflected in an increasingly non-Fickian behavior (Nissan and Berkowitz, 2019). The emergence and intensification of preferential flow paths can be considered a sign of non-equilibrium in the system as they indicate the presence of gradients in parameters such as hydrodynamic pressure and solute concentration (Berkowitz and Zehe, 2020, Zehe et al., 2021, Shavelzon and Edery, 2024). Such system is, therefore, subject to treatment and analysis within the non-equilibrium thermodynamic framework.

# 1.2 Analysis of non-equilibrium thermodynamic systems

The development of non-equilibrium thermodynamics can be attributed to Onsager, 1931, who framed earlier research on irreversible processes and non-equilibrium systems into a unified formulation. A non-equilibrium system is characterized by the presence of gradients in thermodynamic potentials, such as pressure, temperature and chemical potential (also known as thermodynamic forces), that implies a net transfer of energy and matter, denoted as a thermodynamic flux, within the system or across its boundaries. Such processes may be seen as irreversible or dissipative, meaning that the energy, available to perform useful work in the system (also known as the free energy), is being expended, requiring a constant energy supply to the system in order to maintain these fluxes. An important outcome of non-equilibrium thermodynamics is that dissipative processes produce entropy at a rate that is directly related to the power dissipated during the process, the proportionality constant between them being the local temperature (Gouy, 1889). Since the entropy generation reflects the dissipative dynamics of the physical process, or the associated free energy losses, it can be viewed as the governing property for that process (see Hassanizadeh and

Gray, 1990 for applications to transport phenomena). Thus, by studying entropy generation contributions due to dissipative processes pertinent to a non-equilibrium system, important observations can be made regarding the evolution of the system due to internal interactions (Prigogine, 1947).

Based on the work by Onsager, Prigogine proved the *minimum entropy generation* theorem stating that, for an open non-equilibrium system, a stationary state corresponds to a minimum entropy generation rate under the constraint of free energy inflow from the surroundings that prevents the system from achieving equilibrium (this theorem is also known as the minimum energy dissipation theorem). Thus, in a characteristic scenario of flow in porous medium, the free energy inflow into the system is represented by the hydraulic pressure gradient, that can be applied by a pump driving the flow through the medium. According to the minimum entropy generation theorem, minimum rate of energy dissipation in such system corresponds to a stationary flow state, into which the system sets after the transient fluctuations have died out. This approach allows considering the non-equilibrium stationary state as a generalization of the classical concept of equilibrium (Jaynes, 1980).

# 65 1.3 Self-organization as a manifestation of non-equilibrium systems

On many occasions, non-equilibrium systems are accompanied by emergence of self-organization, which can be defined as a broad range of pattern-formation processes, occurring through interactions internal to the system, without intervention by external directing influences (Camazine et al., 2001). Thus, biological morphogenesis during which living organisms acquire their shape and form (Karsenti, 2008), river network formation (Stolum, 1996) and emergence of patterns in autocatalytic chemical reactions (Turing, 1952) are all examples of self-organization in a broad range of physical systems. For subsurface transport, Berkowitz and Zehe, 2020 have suggested that transport in heterogeneous groundwater systems can exhibit certain self-organization traits as well, expressed via spatially correlated, anisotropic patterns of structural and hydraulic properties. Thus, during geochemical weathering, the emergence of preferential flow paths due to reaction-transport interaction, characterized by gradients of pressure and concentration, can be viewed as an expression of self-organization.

75

Self-organization can be characterized quantitatively employing the Shannon entropy, a concept first introduced in information theory by Shannon, 1948. The definition of Shannon entropy is reminiscent of the thermodynamic entropy defined by Gibbs, which quantifies the possible micro states of a system compatible with macroscopic thermodynamic properties that describe the system's macro state (Haken, 1983). Thus, for a large number of gas molecules in a container, a micro state of the system is given by the position and momenta of all molecules, while its macroscopic state is defined by pressure and temperature in the container. According to the second law of thermodynamics, entropy arrives at its maximum in the state of equilibrium, where each micro state is equally likely, corresponding to a homogeneous spatial distribution of gas molecules in a container, while their velocities obey the Maxwell-Boltzmann distribution (Haken, 1983). This homogeneous equilibrium state corresponds to a complete absence of self-organization in the system, with no discernible pattern in arrangement of gas molecules. While the concepts of information theory may seem distant from our discussion, this discipline is intimately connected to the subject of thermodynamics. In fact, it can be shown that the whole of thermodynamics and statistical mechanics can be developed

from purely informational arguments, simplifying and clarifying the underlying formalism and allowing for a more intuitive understanding of the concept of entropy as a measure of uncertainty in the system (Ben-Naim, 2008). An important distinction lies in the broadness of applicability: while the Shannon entropy can defined on any statistical distribution, physical entropy is applicable to a specific set of parameters defined for the specific physical process or system (see Ben-Naim, 2017 and the series of articles on this topic by Arieh Ben-Naim).

# 1.4 Application of non-equilibrium thermodynamics to transport in porous media

In recent years, several studies have applied the thermodynamic perspective to characterize transport self-organization and the emergence of preferential flow paths in flow in porous media. Thus, Hansen et al., 2018 and Hansen et al., 2023 have employed the Jaynes' maximum entropy principle, directly related to the Shannon entropy (Jaynes, 1957), to immiscible and incompressible stationary two-phase flow in a porous medium to determine the most probable flow configuration for that scenario. They have arrived at new variables, analogous to temperature and chemical potential in classical statistical mechanics, as well as a set of Maxwell-like relations between the governing variables of the problem. Hergarten et al., 2014 have proposed to apply the principle of minimum energy dissipation, previously employed to analyze river networks, to preferential flow patterns in subsurface flow systems, and obtained flow patterns that obey this principle for a given recharge under the constraint of a given total porosity. Zehe et al., 2021 have applied Shannon entropy to quantify transport self-organization in artificially generated porous media fields of various degrees of heterogeneity. They have found a clear correlation between the heterogeneity of the medium and the resulting intensity of the preferential flow path phenomenon and gave it an explanation within the thermodynamic framework, showing that a steeper decline in Shannon entropy due to a more pronounced channelization in preferential flow paths can be explained by a higher power of the transversal flow components. Schiavo et al., 2022 have applied the concept of optimal channel networks in the sense of minimum energy dissipation to subsurface flow paths, driven by gravity. They have conducted a statistical Monte Carlo analysis to obtain a probabilistic description of a preferential groundwater network based on limited stratigraphic data available and obtained a good consistency with the known main flow directions and key subsurface flow patterns. Shavelzon and Edery, 2024 have employed Shannon entropy to describe the evolution of preferential flow paths due to dynamic interaction between the transport and chemical reaction in an initially homogeneous porous medium at varying Peclet numbers. They have shown an emergence of transport self-organization in the medium and have correlated its intensity with the Peclet number of the flow. The similarity in the governing principles for the surface and subsurface flows may be remarked in passing (Berkowitz and Zehe, 2020). For application of the concept of minimum energy dissipation to river networks see, for instance, Howard, 1990, Rodriguez-Iturbe et al., 1992, Kleidon et al., 2013.

# 115 1.5 Objectives

100

105

The above studies have drawn a clear connection between transport self-organization, as manifested by the established preferential flow paths in the medium, and entropy generation in the system for the case when the free energy inflow into the system is of a purely hydraulic nature. However, still a need exists for a unified framework able to establish the correlation between the emergence of preferential flow paths and the associated dissipative dynamics of reaction-transport interaction, characteristic of

20 chemical weathering. This problem presents a formidable challenge, since two competing sources of free energy inflow into the system exist in the form of hydraulic pressure gradient that drives the flow through medium and chemical potential gradient due to inflow of reactive species that leads to mixing and reaction.

As a case study, we consider a 2D numerical simulation of reaction-transport interaction on a Darcy-scale, representative of chemical weathering, where the reaction is caused by an injection of a low-pH water in a porous calcite matrix, leading to a reversible dissolution-precipitation reaction with the matrix. The Lagrangian particle tracking approach, modeled after Edery et al., 2021, Shavelzon and Edery, 2024, is employed to simulate the reaction-transport interaction in matrices of varying degree of heterogeneity in transport properties. Non-equilibrium thermodynamic framework to characterize chemical weathering in a porous matrix is developed and employed to analyze the ensuing reaction-transport interaction. We identify the entropy generation sources, attributed to the dissipative processes inherent to this physical scenario and investigate the correlation between the emergence and intensification of preferential flow paths, characterized using Shannon entropy, and the accompanying dissipative dynamics due to flow percolation, mixing of chemical constituents and reaction. In Section 2 the methodology of the reactive transport simulation is presented, followed by the presentation of the fundamentals of non-equilibrium thermodynamics and their application to chemical weathering in porous media in Section 3. The obtained results are presented and discussed in Section 4. Summary and conclusions complete the manuscript.

# 2 Methodology: Modeling the reaction-transport interaction, representative of chemical weathering

To facilitate the description of our research procedure, we provide the reader with the conceptual flowchart for methodology, presented in Figure 1. Each graphical block, shown by different colors in the flowchart, references the specific section in the paper where it is explained in details. Thus, Chapter 2 describes Blocks 1 and 2 (see also Supplementary 1 for algorithm validation) and Chapter 3 explains Blocks 3 and 4 (see also Supplementary 2 for background on non-equilibrium thermodynamics) References to this flowchart will be given throughout Chapters 2 and 3 in the appropriate places.

# 2.1 Flow and transport in porous matrix

140

The dynamics of the reaction-transport interaction in a heterogeneous porous calcite matrix is investigated using a 2D computational field that represents a rectangular sample of calcite mineral of dimensions  $L_x \times L_y$ , initially in chemical equilibrium with the resident water of pH 8. Low-pH water (pH of 3.5) is injected at the inlet, disrupting the existing chemical equilibrium and reacting with the porous calcite matrix. The computational field is represented by the hydraulic conductivity distribution K and is discretized into  $150 \times 60$  computational cells, each having the dimensions  $\Delta x, \Delta y$  (here, x and y are axes in the direction of the flow and transverse to it, respectively).

Hydraulic conductivity fields of varying degrees of heterogeneity, defined by the prescribed values of conductivity variance  $\sigma_0^2$  and spatial correlation length  $l_c$ , are generated using the SGSIM sequential Gaussian simulator (Deutsch and Journel, 1997),

Figure 1. Flowchart for the research procedure.

while the obtained realizations are considered as log-conductivity distributions. Varying  $\sigma_0^2$  values are considered, while  $l_c$  is kept constant such that the ratio  $\Delta x/l_c=4$  is obtained (the subscript 0 denotes parameters referring to the initial hydraulic conductivity distribution, prior to the occurrence of reaction). This is considered a good enough discretization to capture the bigger-scale structures in the heterogeneous field and their effect on the statistics of the coupled transport-reactive process. This procedure corresponds to Block 1 in Figure 1. Constant initial porosity value of  $\theta_0=0.43$  is assumed for each field realization. This assumption, introduced in order to facilitate the chemical aspects of our reactive transport model, was deemed acceptable, as the purpose of our model is to capture the qualitative dynamics of the complex process of geochemical weathering, thus making it possible to simulate and analyze the complex reversible behavior using a relatively unsophisticated model (see Section 2.3 in Shavelzon and Edery, 2024 for details).

Flow through the calcite porous matrix is obtained by applying a hydraulic pressure head drop boundary condition  $\Delta h_{BC}$  between the inlet (left boundary) and outlet (right boundary) of the field, while the horizontal boundaries represent impenetrable walls. The distributions of the hydraulic pressure head h and the water Darcy flux  $\mathbf{q}$  in the computational field as functions

of the spatial coordinate **x** are governed by the continuity equation and the Darcy's law (1)–(2), solved using Finite element method (Guadagnini and Neuman, 1999). They are then used to calculate the flow velocity field **v**, required to simulate the reactive transport in the matrix. Throughout the paper, vector fields will be specified using bold font, while scalar fields will appear in the regular font.

$$\nabla \cdot \mathbf{q}(\mathbf{x}) = 0 \tag{1}$$

$$\mathbf{q}(\mathbf{x}) = -K\nabla h(\mathbf{x})$$
 (2)

We employ the Lagrangian particle tracking approach to simulate the reactive solute transport in the porous matrix (Borgne et al., 2008). The invading low-pH water is represented by the reactive particles injected at a constant rate at the left upstream boundary, flux-weighted according to the conductivity distribution of the inlet cells. The Langevin equation, which combines the deterministic contribution of the advective flow with the stochastic diffusive process, is employed to advance the injected particles in the matrix. The general Langevin equation for a stochastic vectorial variable **X** has the form (Risken, 1996)

$$\frac{d\mathbf{X}}{dt} = \mathbf{h}(\mathbf{X}, t) + g(\mathbf{X}, t)\mathbf{\Gamma}(t)$$
(3)

where  $h(\mathbf{X},t),g(\mathbf{X},t)$  are functions of  $\mathbf{X}$  and the computational time t (not to confuse with the hydraulic head distribution h), and  $\Gamma(t)$  is a vectorial Gaussian random variable with independent and standard-normally distributed components. To simulate the trajectory of a solute particle moving in the porous calcite matrix,  $\mathbf{X}(t)$ , affected by both the advective velocity field  $\mathbf{v}$  and the diffusive transport process with diffusion coefficient D, we set  $\mathbf{h} = \mathbf{v}, g = \sqrt{2D}$  (Perez et al., 2019). Here,  $\mathbf{X}(t)$  is the position of a particle,  $\mathbf{v}$  is the flow velocity field, obtained from (1)–(2), t is the computational time and D is the diffusion coefficient of the injected fluid in the matrix. For the diffusion coefficient, a representative value of D = 1e-5 [cm<sup>2</sup> min<sup>-1</sup>] was employed (Domenico and Schwartz, 1997). To be solved numerically, we discretize (3) using the Euler-Maruyama method (Kloeden, 1992). We employ the ran2 random Gaussian generator (Press et al., 1992) for the stochastic variable generation. Determination of flow and transport in the matrix, as described above, corresponds to the first three steps of Block 2 in Figure 1.

The Lagrangian particle tracking simulation has been validated using the scenario of a 2D instantaneous injection in a homogeneous porous medium, governed by a steady state flow. For validation, we used the equivalence property of a statistical ensemble governed by the Langevin equation (3) to the advection-diffusion equation (Risken, 1996, Perez et al., 2019), whose solution for this scenario constitutes a traveling Gaussian wave (Kreft and Zuber, 1978). See Supplementary 1 in the manuscript for validation results.

# 2.2 Chemical reaction model

Upon entering the calcite matrix, initially saturated and in chemical equilibrium with the resident water, the injected low-pH water disrupts the existing equilibrium and causes dissolution of calcite along with the production of dissolved calcium and

195 carbonic acid, as shown by the following chain of reactions (Singurindy and Berkowitz, 2003, Edery et al., 2011).

$$HCO_3^- + H^+ \leftrightarrow H_2CO_3$$
 (4)

$$CO_3^{2-} + H^+ \leftrightarrow HCO_3^- \tag{5}$$

$$CaCO_{3(s)} \leftrightarrow CO_3^{2-} + Ca^{2+} \tag{6}$$

Here, hydrogen ions  $H^+$ , associated with the injected low-pH water, combine with  $CO_3^{2-}$  to produce  $H_2CO_3$ , thus causing dissolution of the calcite porous matrix in accordance with the Le Chatelier principle. The pH level inside the matrix is assumed to be bounded by that of the invading (pH 3.5) and the resident fluid (pH 8). This simplified chemical representation can be seen as a particular instance of Edery et al., 2011 that also includes de-dolomitization, and was treated extensively in Edery et al., 2021 and Shavelzon and Edery, 2024 (see also Singurindy and Berkowitz, 2003, Al-Khulaifi et al., 2019).

In our model, the injected low-pH water is associated with a source of both  $Ca^{2+}$  and  $H^+$ . We assume an abundance of dissolved  $Ca^{2+}$ , thus making the reaction (6) non-rate-limiting (this assumption is not that far-fetched in karst systems - see Romanov et al., 2003, Eang et al., 2018). Therefore, reactions (4)-(5) become the rate-limiting factor, and the reactive process is governed by the available  $H^+$ , in consistency with observations from previous studies (Edery et al., 2011, Singurindy and Berkowitz, 2003). The spatial distribution of  $H^+$  in the calcite matrix is, in turn, controlled by the transport processes. Combining reactions (4)-(6) along the lines of Edery et al., 2021 and Shavelzon and Edery, 2024, we obtain the following simplified equation that represents an interaction between hydrogen ions, carbonic acid and calcite

$$a = h + c \tag{7}$$

where a, h, c denote  $H_2CO_3$ ,  $2H^+$ ,  $CaCO_3$ , respectively. While simplified, this chemical reaction setup nevertheless captures the qualitative dynamics of chemical weathering, making it possible to analyze the complex reversible behavior using a relatively unsophisticated model. The dissolution-precipitation reaction is reversible, with precipitation possible under favorable pH conditions. The direction of reaction is governed by the departure from chemical equilibrium, characterized by concentrations of  $H^+$  and  $H_2CO_3$  in the area of reaction: dissolution will occur for  $H_2CO_3$  concentration above the equilibrium value for the current pH and vice versa, based on the Bjerrum plot for carbonic acid dissociation (Supplements in Edery et al., 2021, Manahan, 2000). For a detailed description of the computational setup, refer to Edery et al., 2021 and Shavelzon and Edery, 2024.

# 2.3 Chemical kinetics and reaction-transport interaction

In the presented reactive transport model, the kinetics of the reactive process are developed to mimic the actual chemical weathering process that take place in practice. At t = 0,  $H^+$  particles, that represent the low-pH water, are being injected into the matrix at a constant rate and advanced in accordance with (3) for the duration of the computational time step  $\Delta t$ . Then, reaction occurs in a cellwise fashion as described by (7), during which some of the  $H^+$  particles in each computational cell transform into  $H_2CO_3$  particles or vice versa, accompanied by dissolution or precipitation of calcite. The direction of reaction

is determined by the deviation from chemical equilibrium in a specific cell (during the first reaction event only dissolution takes place, since no  $H_2CO_3$  particles are present in the matrix initially). Each  $H^+$  particle represents a molar parcel of hydrogen ions. According to (7), each newly created  $H_2CO_3$  particle receives half of this amount of carbonic acid (see Shavelzon and Edery, 2024 for the exact procedure of determining the parcel size). We assume that the characteristic reaction timescale is negligible with respect to the transport timescale, thus the reaction is instantaneous and proceeds until chemical equilibrium is reached everywhere in the field. This corresponds to an infinite local Damkohler number.

To determine the amount of dissolved or precipitated calcite in each particular cell, the amount of  $H^+$  and  $H_2CO_3$  there is assessed based on the current number of particles of both types in the cell. The current pH level in a cell is estimated based on the number of  $H^+$  particles there and is employed to calculate the fractional amount of  $H_2CO_3$  in equilibrium  $\alpha_1(pH)$  as

$$\alpha_1(pH) = \frac{10^{-2pH}}{10^{-2pH} + 10^{-pk_{a1}} + 10^{-pk_{a2}} + 10^{-pH - pk_{a1}}}$$
(8)

with  $pk_{a1}=6.35, pk_{a2}=10.33$  (see Manahan, 2000 and Supplementary in Edery et al., 2021 for related data and the Bjerrum plot for carbonic acid dissociation). We then compare the value of  $\alpha_1(pH)$  to the actual fractional amount of  $H_2CO_3$  in a cell (see Section 2.3 in Shavelzon and Edery, 2024 for details). The reaction proceeds in the direction, established corresponding to the Le Chatelier principle for (7): we expect dissolution in case when the actual fractional amount of  $H_2CO_3$  is smaller than its value in equilibrium  $\alpha_1(pH)$ , and vice versa. Equilibration in a cell consists of a series of reactive events, during each of which a single  $H^+$  particle transforms into a  $H_2CO_3$  particle or vice versa. Current values of pH level and the fractional amount of  $H_2CO_3$  in a cell are recalculated following each event and the process is repeated until equilibrium in a cell is reached, so that the actual and the equilibrium values of the  $H_2CO_3$  fractional amount are close enough. In the current simulation, an error of 5 % in establishing the chemical equilibrium in a cell was considered acceptable.

According to (7), the change in cell volume due to dissolution or precipitation of calcite there is equal to the change in the molar amount of  $H_2CO_3$  due to reaction in the cell. Upon establishing equilibrium, we update the cell porosity  $\theta$  as shown in (9), where the subscripts k+1 and k denote values at the current and previous computational steps, respectively (after and prior to the reaction),  $d[H_2CO_3]$  is the total increment of carbonic acid in the cell due to equilibration in the current time step (accepts negative values as well),  $M_{CaCO_3} = 36.93 \, [\text{cm}^3 \, \text{mol}^{-1}]$  is the molar volume of calcite (Morse and Mackenzie, 1990) and  $\Delta x \Delta y$  is the cell volume assuming unit depth. We then employ the Kozeny-Carman relation (10) to update the hydraulic conductivity K value in a cell

$$\theta_{k+1} = \theta_k + \frac{d[H_2CO_3]M_{CaCO_3}}{\Delta x \Delta y} \tag{9}$$

$$K_{k+1} = K_k \cdot \frac{\theta_{k+1}^3}{(1 - \theta_{k+1})^2} \cdot \frac{(1 - \theta_k)^2}{\theta_k^3} \tag{10}$$

This process is repeated for all computational cells before injecting new particles and advancing all existing particles again during the next time step  $\Delta t$ . The hydraulic pressure head distribution h is recalculated for the reaction-altered hydraulic

conductivity field from (1)–(2) at constant time intervals of  $10\Delta t$ , thus introducing the reaction-transport interaction in the computational model. Chemical equilibration and the ensuing update of the porosity and hydraulic conductivity distributions, as described above, correspond to the last two steps of Block 2 in Figure 1.

Reaction enhancement is employed in the simulation by magnifying the cell porosity increment due to reaction (9) by a certain factor, thus accelerating the reaction-transport interaction in the field, while leaving unamended the overall reaction-transport dynamics. This allows reduction of computational costs (Shavelzon and Edery (2024)). Distributions of porosity and hydraulic conductivity, pressure head and Darcy flux,  $H^+$  and  $H_2CO_3$  concentrations and the statistics of dissolution-precipitation reactive events are recorded each  $10\Delta t$  units of time to be used in the subsequent analysis within the non-equilibrium thermodynamic framework. The numerical simulation proceeds until the time equal to twice the pore volume time  $T_{pv}$  is reached.  $T_{pv}$  is determined for each SGSIM-generated realization of hydraulic conductivity K using the non-reactive particle tracking algorithm (NRPT), where the particles are being injected at the inlet and traverse the field, subject to the applied hydraulic head drop  $\Delta h_{BC}$ , while omitting the reaction part, and is set to be the time when 50% of the injected particles have reached the outlet.

# 3 Establishing the correlation between the intensification of preferential flowpaths and dissipative dynamics during chemical weathering

Upon completing the chemical weathering simulation as described in Chapter 2 (Block 2 in Figure 1), we employ the obtained results to investigate the correlation between the emergence and intensification of preferential flowpaths and the evolution of dissipative dynamics due to reaction-transport interaction in our system. We analyze the latter using tools from non-equilibrium thermodynamics (Block 3 in Figure 1), while quantification of the former requires basics of information theory (Block 4 in Figure 1). This Chapter presents the methodlogy for establishing this correlation, including basic concepts and results for both disciplines.

# 3.1 Non-equilibrium thermodynamics for chemical weathering

Non-equilibrium in a system is essentially reflected by the presence of gradients in *thermodynamic potentials* such as pressure, temperature and chemical potential (also known as *thermodynamic forces*). These forces drive *thermodynamic fluxes*, such as fluxes of energy or matter, within the system or between the system and its surroundings. Processes that involve transfer of energy or matter against potential gradient may be seen as *dissipative*, since the *free* energy in the system (energy available to perform useful work) is being expended during such process. Thus, energy must be constantly supplied in order to maintain these fluxes and the state of non-equilibrium in the system.

When applying the non-equilibrium thermodynamic framework to the problem of reactive flow in porous media on a Darcy scale, characteristic of chemical weathering, our system can be viewed as subject to free energy inflow in the form of (a)

hydraulic head gradient that maintains flow through the medium and (b) inflow of reactive species which creates chemical potential gradient in the porous medium, leading to mixing and reaction. This energy is being spent via the dissipative dynamics, created by the following interrelated processes: (a) *percolation*, which in the context of our work corresponds to fluid transport through the porous matrix; this involves dissipation of hydraulic power while overcoming the hydraulic resistance of the matrix due to viscous friction effects, affecting concentration distribution of the chemical species and directly influencing chemical reaction, (b) *mixing* of chemical constituents that involves dissipation of chemical energy and affects the reaction rate and (c) *chemical reaction* of dissolution-precipitation, also involving dissipation of chemical energy, which affects directly the transport through matrix by altering the transport properties of the matrix. This is depicted graphically in Figure 2 that presents the schematics of energy inflow and dissipation mechanisms in chemical weathering. We neglect temperature variations and ensuing heat transfer effects in the porous matrix. Thus, in our model problem the conjugate pairs of thermodynamic potential gradients that drive the corresponding fluxes are: (a) hydraulic head gradient that drives the mass flux, (b) chemical potential gradient that drives the molar fluxes of the chemical species and (c) gradient of the partial molar Gibbs energy that drives the extent of reaction.

An important outcome of non-equilibrium thermodynamics is that dissipative irreversible processes produce entropy at a

Figure 2. Sources of inflow and dissipation of free energy in subsurface chemical weathering.

rate that is directly related to the power dissipated during the process, the proportionality constant between them being the local temperature (Gouy, 1889). This allows making important observations regarding the evolution of a non-equilibrium system

due to interactions within, as well as with surroundings, by studying entropy generation due to pertinent dissipative processes.

In the remainder of the section, the non-equilibrium thermodynamic framework for chemical weathering will be presented that
accounts for the above described sources of inflow and dissipation of free energy. Supplementary 2 describes the basic concepts
of non-equilibrium thermodynamics. For an in-depth examination of the subject, see Kondepudi and Prigogine, 1998.

We consider a general scenario of a multi-component flow of miscible species in porous media such as rock or soil, where processes are driven by temperature-, pressure- and chemical composition gradients. Our system consists of n components, where the first n-1 are the species that participate in the flow and the porous solid matrix is the n-th, among which chemical interactions are possible. The chemical interactions are represented by the total of r chemical reactions, written in the general form as

$$\sum_{i,i} \nu_{ji} c_i = 0 \tag{11}$$

where j is the reaction number,  $c_i$  is the molar concentration of the i-th component in the flow and  $\nu_{ji}$  is the stoichiometric coefficient of the i-th component in the j-th reaction, its extent given by  $d\xi_j/dt$ . The balance equation for the molar concentration of component i is given by

$$\frac{\partial c_i}{\partial t} = -\nabla \cdot \mathbf{J_i} + \sum_{i} \nu_{ji} \frac{d\xi_j}{dt}$$
(12)

where  $J_i$  is the molar flux of the *i*-th component, given in the laboratory reference frame (Kjelstrup and Bedeaux, 2008, De Groot and Mazur, 1984). Thus, the rate of change of the concentration of the *i*-th component in a local volume element is affected by the net flow into the volume and production by chemical reactions. The balance equation for the internal energy density u, under the simplifying assumption of mechanical equilibrium, is given by

$$\frac{\partial u}{\partial t} = -\nabla \cdot \mathbf{J}_u \tag{13}$$

where the internal energy flux  $\mathbf{J}_u = \mathbf{J}_q + \sum_i h_i \mathbf{J}_i$  is given by the measurable heat flux  $\mathbf{J}_q$  and the contributions of the chemical components,  $h_i$  being the specific enthalpy of the *i*-th component. The mechanical equilibrium assumption neglects energy dissipation due to viscosity-related velocity gradients, pressure gradients, and body forces. This assumption is often valid for more general cases and in our case may be justified by the low pressure gradients and velocities that are typical for flow in porous media (see Kjelstrup et al., 2018 and a general discussion in De Groot and Mazur, 1984). The internal energy of the porous matrix is constant and its chemical composition unchanged, thus its chemical potential is constant as well. Writing the balance equation for the entropy density s in a similar way, we obtain

$$\frac{\partial s}{\partial t} = -\nabla \cdot \mathbf{J}_s + \sigma \tag{14}$$

where  $J_s$  is the entropy flux, given in the laboratory reference frame and  $\sigma \ge 0$  is the entropy generation term that accounts for the irreversibility of the process. Notice that the change in the entropy density is the sum of the divergence of the entropy flux (a conservative term) and the entropy generation (a source term which determines the dissipation of the free energy). Assuming that thermodynamic equilibrium still holds locally within small system regions, despite interactions between different

regions and with surroundings, allows defining the intensive thermodynamic variables such as temperature T, pressure p, and chemical potential  $\mu$  and employ the usual thermodynamic relations, such as the Gibbs relation, on a local scale (Kondepudi and Prigogine, 1998). Thus, we can write the Gibbs relation for an infinitesimal element during the time interval dt as

$$\frac{\partial s}{\partial t} = \frac{1}{T} \frac{\partial u}{\partial t} - \frac{1}{T} \sum_{i} \mu_{i} \frac{\partial c_{i}}{\partial t}$$

$$\tag{15}$$

where  $\mu_i = h_i - Ts_i$  is the chemical potential of species i and  $h_i$ ,  $s_i$  are the corresponding specific enthalpy and entropy. 345 Inserting the expressions (12)–(13) into (15), together with the definition of the internal energy flux  $\mathbf{J}_u$  given above, and rearranging in the form of (14), we obtain

$$\frac{\partial s}{\partial t} = -\nabla \cdot (\frac{\mathbf{J}_u - \sum_i \mu_i \mathbf{J}_i}{T}) + \nabla (\frac{1}{T}) \cdot \mathbf{J}_u - \sum_i \nabla (\frac{\mu_i}{T}) \cdot \mathbf{J}_i - \sum_j \frac{\Delta g_j}{T} \frac{d\xi_j}{dt}$$
(16)

where  $\Delta g_j = \sum_i \nu_{ji} \mu_i$  is the partial molar Gibbs energy of the j-th reaction. By comparing (16) and (14), the entropy flux and the entropy generation term can be identified as

$$\mathbf{J}_s = \frac{1}{T} (\mathbf{J}_u - \sum_i \mu_i \mathbf{J}_i) \tag{17}$$

$$\sigma = \nabla(\frac{1}{T}) \cdot \mathbf{J}_u - \sum_i \nabla(\frac{\mu_i}{T}) \cdot \mathbf{J}_i - \sum_j \frac{\Delta g_j}{T} \frac{d\xi_j}{dt}$$
(18)

Noting that  $\mu_i = \mu_i(T, p, c_i)$  and rewriting the chemical potential gradient as

$$\nabla \mu_i = -s_i \nabla T + v_i \nabla p + (\nabla \mu_i)_{T,p} \tag{19}$$

where  $(\nabla \mu_i)_{T,p} = \sum_j \frac{\partial \mu_i}{\partial c_j} \nabla c_j$  represents the chemical potential gradient at constant T, p and  $s_i = -(\frac{\partial \mu_i}{\partial T})_{p,c_i}$ ,  $v_i = (\frac{\partial \mu_i}{\partial p})_{T,c_i}$  are the specific entropy and molar volume (reciprocal of molar density) of the *i*-th component, we obtain for (18)

$$\sigma = \nabla (\frac{1}{T}) \cdot \mathbf{J}_q - \sum_{i} \frac{(\nabla \mu_i)_{T,p}}{T} \cdot \mathbf{J}_i - \frac{\nabla p}{T} \cdot \sum_{i} v_i \mathbf{J}_i - \sum_{i} \frac{\Delta g_j}{T} \frac{d\xi_j}{dt}$$
(20)

Notice the bilinearity of entropy generation  $\sigma$  in terms of thermodynamic forces and fluxes, which allows writing the general form of the entropy generation term as

$$\sigma = \sum_{i} \mathbf{F}_{i} \cdot \mathbf{J}_{i} \tag{21}$$

where  $\mathbf{F}_i$ ,  $\mathbf{J}_i$  are the thermodynamic force and flux corresponding to the *i*-th entropy generation source and the index *i* runs on different sources of entropy generation (Kondepudi and Prigogine, 1998). Thus, we are able to identify these forces and the fluxes they drive. The first term in (20) represents entropy generation due to heat transfer, where the thermodynamic force  $\nabla(\frac{1}{T})$  drives the heat flow  $\mathbf{J}_q$ ; the second term accounts for entropy generated due to mixing of chemical species where the force  $\frac{(\nabla \mu_i)_{T,p}}{T}$  drives the molar fluxes  $\mathbf{J}_i$ ; the third term represents entropy generation due to hydraulic energy losses in flow as it overcomes the hydraulic resistance of the porous medium, where the force  $\frac{\nabla p}{T}$  drives the flux  $\sum_i v_i \mathbf{J}_i$ ; finally, the last term represents entropy generation due to chemical reaction, where the force  $\frac{\Delta g_j}{T}$  drives the *j*-th reaction extent  $\frac{d\xi_j}{dt}$ . Throughout the paper we will assume negligible temperature variations, thus the first term in  $\sigma$  will be neglected. We will denote the last

three remaining entropy generation mechanisms as *mixing*, *percolative* and *reactive* entropy generation rates. Calculation of the entropy generation terms, as described above, correspond to Block 3 in Figure 1. For detailed procedure to calculate the entropy generation terms in our model, see Supplementary 3.

# 3.2 Employing Shannon entropy to quantify transport self-organization

While the dissipative dynamics of chemical weathering can be quantified using non-equilibrium thermodynamics, the emergence and intensification of preferential flow paths due to reaction-transport interaction must be addressed from a different perspective. Noting that the emerging preferential flow paths phenomenon in subsurface flows can be considered as a manifestation of transport self-organization (Berkowitz and Zehe, 2020), we will characterize it employing the Shannon entropy metric, introduced originally in information theory by Shannon, 1948. While analyzing the statistical process of communication, Shannon employed an entropy-like metric to quantify information represented by a series of symbols, that constitute a transmitted message.

To quantify the evolution of transport self-organization due to reaction-transport interaction during chemical weathering, we employ the Shannon entropy along the lines of Zehe et al., 2021 and Shavelzon and Edery, 2024, where it was used in the context of subsurface flow in heterogeneous porous medium and in the dissolution-precipitation setting in an initially homogeneous porous matrix. Based on the hypothesis by Berkowitz and Zehe, 2020 that have correlated transport self-organization with the intensification of concentration gradients in the direction normal to flow, we write the Shannon entropy metric for transport self-organization *S* as

$$S(x,t) = -\Sigma_i p_i \log_2 p_i \tag{22}$$

where S(x,t) is the Shannon entropy of the transversal solute concentration of a specific chemical species (either  $H^+$  or  $H_2CO_3$  in our model) at a given axial coordinate x and a given time t,  $p_i = N_i/N_{tot}$ ,  $i = 1...N_y$  is the relative concentration of that species at x,t in the direction transverse to flow,  $N_i$  is the current amount of particles that represent the species in the i-th cell at x and  $N_{tot}$  is the current total number of particles of this species in all cells at x.

Shannon entropy (22) obtains its maximum value for a uniform species concentration in the direction transversely to flow, when each flow path through the domain is equally likely (the state of a uniform concentration distribution can be viewed as reflecting the state of equilibrium in classical thermodynamics). In that case, it is easily shown that the Shannon entropy value depends only on the discretization in the y-direction:  $S(x,t) = \log_2 N_y$ . Deviations from this maximum value indicate emergence of transport self-organization, suggesting that some paths are more preferable for the flow than the others. These preferential flow paths are characterized by an elevated solute concentration, compared to the mean value. In the extreme case of a single path that funnels all of the solute, the Shannon entropy obtains its minimum value of zero (Zehe et al., 2021, Shavelzon and Edery, 2024). Calculation of the Shannon entropy of transport self-organization, as described above, correspond to

400 Block 4 in Figure 1. For a short account of the Shannon entropy metric, refer to the Supplementary data in Shavelzon and Edery, 2024.

# 4 Results and discussion

We simulate the evolution of chemical weathering in calcite porous matrix using SGSIM-generated hydraulic conductivity fields with  $\sigma_0^2 = 1,3,5$ , where  $\sigma_0^2$  is the variance of the initial conductivity distribution. A total of 20 realizations of hydraulic conductivity fields were created for each value of  $\sigma_0^2$  to form a statistical ensemble of a reasonable size. For each field realization, we simulate the reactive transport scenario by applying a constant value of the inlet-outlet hydraulic pressure head boundary condition  $\Delta h_{BC} = 100 [\mathrm{cm}]$  and running the reactive particle tracking simulation as described in Section 2. We inject 2e5  $H^+$  particles per pore volume time  $T_{pv}$ , while the molar parcel assigned to each particle equals to 2.28e-10 [mol] of  $H^+$ , based on considerations described in Section 2.3 of Shavelzon and Edery, 2024. We employ a reaction enhancement by a factor of 3e2 (see Section 2.3 in the paper). The numerical simulation is terminated at the computational time of  $2T_{pv}$ . Unless specified otherwise, all plots present an average of 20 realizations for each  $\sigma_0^2$ .

We begin the section by analyzing the traits in the spatial distribution of reactants and dissolution-precipitation reactions in the porous matrix, investigate their effect on the emergence and evolution of preferential flow paths, continue by analyzing the dissipative dynamics of chemical weathering and look into the correlation between the two. We also present implications of our study for the subject of subsurface carbon sequestration by mineralization.

# 4.1 Spatial distribution of reactants and dissolution-precipitation reactions

To understand the origins of emergence of preferential flow paths during chemical weathering of a calcite matrix, we first examine the overall traits in the spatial distribution of reactants and dissolution-precipitation reactions. Figure 3 presents the initial hydraulic conductivity field  $K_0$  for  $\sigma_0^2=1,3,5$  (frames (a)–(c)) and the increment in hydraulic conductivity due to reaction-transport interaction relative to the initial field  $K-K_0$  at a dimensionless time  $\tilde{t}=1.0$ , defined as  $\tilde{t}=t/T_{pv}$ , for  $\sigma_0^2=1,3,5$  (frames (d)–(f)) - a single realization of the reactive process. Here,  $\tilde{x}=x/L_x$  and  $\tilde{y}=y/L_y$  are the normalized coordinate axes. Note that the color code range in frames (a)–(c) differs for better visualization. In frames (d)–(f), flow streamlines are denoted by red curves. By observing the frames (a)–(c) in Figure 3, the influence of  $\sigma_0^2$  on the hydraulic conductivity distribution in the fields becomes evident, as higher  $\sigma_0^2$  values clearly correspond to more heterogeneous fields. From the hydraulic conductivity increment plots (frames (d)–(f)), we conclude that reaction mostly follows the path prescribed by the flow streamlines, coinciding at least partially with the high conductivity regions. A significant portion of reactions occur in the areas where flow streamlines are the most dense, indicating a higher flow rate. This can be explained by the fact that these highly conductive areas attract a significant portion of reactive species, thus facilitating the reaction there. The influence of the established preferential flow paths becomes more pronounced with the increasing  $\sigma_0^2$ , as the reaction areas become more localized and intense (Shavelzon and Edery, 2024). Near the inlet area, the chemical equilibrium is inclined in the direction of dissolution due to the

Figure 3. Reaction-transport interaction in fields of varying heterogeneity: Initial hydraulic conductivity field  $K_0$  for (a)  $\sigma_0^2 = 1$ , (b)  $\sigma_0^2 = 3$ , (c)  $\sigma_0^2 = 5$ , and Increment in hydraulic conductivity due to reaction-transport interaction relative to the initial field  $K - K_0$  at a dimensionless time  $\tilde{t} = 1.0$  for (e)  $\sigma_0^2 = 1$ , (f)  $\sigma_0^2 = 3$ , (g)  $\sigma_0^2 = 5$  (a single realization of the reactive process). Flow streamlines in the field are denoted by red curves.

constant income of  $H^+$  there. This finding is in agreement with experimental data (Poonoosamy et al., 2020, Deng et al., 2022).

Figure 4 presents the decimal logarithm of the normalized concentration  $\tilde{c}$  of reactive species for  $\sigma_0^2 = 1, 3, 5$  at  $\tilde{t} = 1.0$ , with  $H^+$  species shown in frames (a)–(c) and  $H_2CO_3$  species shown in frames (d)–(f) - a single realization of the reactive process. The normalized concentration of  $\tilde{c}$  is defined as the total molar amount of a species in a computational cell divided by the molar parcel associated with a single particle of that specie. Frames (b) and (e) show close ups of characteristic mean and high conductivity regions for  $\sigma_0^2 = 3$ . It is evident that  $H^+$  and  $H_2CO_3$  spatial distributions follow each other closely as the reactive process evolves. Here, as well, the influence of preferential flow paths becomes more pronounced with the increasing  $\sigma_0^2$ , as the reactive species are being funneled into just a few dominant flow paths, while a significant portion of the field in the periphery does not experience their presence at all (see also Dagan and Edery, 2022). By observing the characteristic color ranges that correspond to the spatial distributions of reactive species for the case of  $\sigma_0^2 = 3$ , it is evident that different species follow somewhat different traits: although the amount of  $H^+$  in the field should quickly exceed the amount of  $H_2CO_3$  due to constant influx of the former at the inlet, we observe that the average normalized concentration of  $H^+$  (dominated by a bright green,  $\tilde{c} \approx 1.1$ ) is somewhat lower than that of  $H_2CO_3$  (dominated by a yellow,  $\tilde{c} \approx 1.3$ ). However, while the color range of the  $H_2CO_3$  concentration distribution suggests a moderate scatter in concentration, signifying a more homogeneous spatial distribution, the color range of the  $H^+$  concentration exhibits a much more significant scatter. We interpret these differences as a direct influence of the established preferential flow paths: due to the constant influx of  $H^+$  at the inlet, its significant portion that travels in these paths exhibits itself by a relatively high concentration (saturated red in the plots). On the other hand, in our

**Figure 4.** Decimal logarithm of the normalized concentration  $\tilde{c}$  of reactive species in fields of varying heterogeneity at  $\tilde{t}=1.0$ :  $H^+$  species for (a)  $\sigma_0^2=1$ , (b)  $\sigma_0^2=3$ , (c)  $\sigma_0^2=5$ , and  $H_2CO_3$  species for (d)  $\sigma_0^2=1$ , (e)  $\sigma_0^2=3$ , (f)  $\sigma_0^2=5$  (a single realization of the reactive process). Frames (b) and (e) show close ups of characteristic mean and high conductivity regions for  $\sigma_0^2=3$ .

model  $H_2CO_3$  is introduced into the field only as a result of dissolution reaction; since there is no constant influx of  $H_2CO_3$ , its concentration is more prone to dispersion, making its spatial distribution more homogeneous.

To quantify the characteristic traits for the spatial distribution of reactants, as well as to assess the intensification of preferential flow paths during the evolution of reaction-transport interaction, we calculated the Shannon entropy of spatial distribution of  $H^+$ ,  $H_2CO_3$  species in the direction transverse to flow as a function of normalized distance from inlet  $\tilde{x} = x/L_x$  and dimensionless time  $\tilde{t}$ ,  $S(\tilde{x},\tilde{t})$ , for different  $\sigma_0^2$  values. We normalized the Shannon entropy as  $S_{norm}(\tilde{x},\tilde{t}) = (S(\tilde{x},\tilde{t}) - S_{max})/S_{max}$ , where  $S_{max} = \log_2 N_y$  is the maximum possible Shannon entropy value for our numerical discretization (see Section 3.2 in the paper). The normalized Shannon entropy  $S_{norm}(\tilde{x},\tilde{t})$  is bounded by the maximum value of 0, which corresponds to a complete lack of self-organization (uniform concentration distribution over the field), and the minimum value of -1, which signifies maximum self-organization possible in the form of a single path which all solute particles follow (Shavelzon and Edery, 2024, Zehe et al., 2021).

Figures 5(a)–(b) present the normalized Shannon entropy of spatial distributions of  $H^+$ ,  $H_2CO_3$  for  $\tilde{t}=2.0$  as a function of  $\tilde{x}$  for  $\sigma_0^2=1,3,5$ . We observe that the normalized Shannon entropy of both species decreases with an increase in  $\sigma_0^2$ , indicating an intensifying transport self-organization within the established preferential flow paths for more heterogeneous fields. Another interesting feature is the decline of the Shannon entropy with the distance from inlet. These results fit well with those reported in Zehe et al., 2021 and Shavelzon and Edery, 2024, where a direct correlation between field heterogeneity and transport self-organization was reported. The Shannon entropy of  $H^+$  is consistently lower than that of  $H_2CO_3$ ; this finding further

Figure 5. Normalized Shannon entropy of spatial distribution of reactive species  $S_{norm}(\tilde{x},\tilde{t}) = (S(\tilde{x},\tilde{t}) - S_{max})/S_{max}$  at  $\tilde{t} = 2.0$  as a function of the normalized distance from inlet  $\tilde{x}$ : (a)  $H^+$  species, (b)  $H_2CO_3$  species, and Normalized mean Shannon entropy of spatial distribution of reactive species over the field  $\tilde{S}_{norm}(\tilde{t}) = (\tilde{S}(\tilde{t}) - S_{max})/S_{max}$  as function of dimensionless time  $\tilde{t}$ : (c)  $H^+$  species, (d)  $H_2CO_3$  species.

supports the hypothesis of different traits in their behavior, indicating that  $H^+$  is more associated with the preferential flow paths, thus exhibiting higher internal organization in the field.

Next, we calculated the *mean* value of Shannon entropy of reactive species over the field as a function of dimensionless time  $\tilde{t}$ ,  $\tilde{S}(\tilde{t})$ , and normalized it as presented above. Figures 5(c)–(d) present the normalized *mean* Shannon entropy of spatial distributions of  $H^+$ ,  $H_2CO_3$  as a function of  $\tilde{t}$ ,  $\tilde{S}_{norm}(\tilde{t})$ , for  $\sigma_0^2=1,3,5$ . We observe that the normalized *mean* Shannon entropy first increases monotonously and then declines at a relatively slow pace. This localized tipping point ( $\tilde{t}\approx 1.0,0.8,0.6$  for  $\sigma_0^2=1,3,5$  respectively) is identified as a point at which preferential flow paths have been established in the field. Before this point is reached, reactants build up in the field, corresponding to an increase in the mean Shannon entropy (as the reactants advance further, covering more area, the overall mean entropy value over the field increases). After this tipping point, the reactants further self-organize within the established paths, as shown by the decreasing Shannon entropy. Here, too, an

additional indication to the intensification of preferential flow paths with an increase in heterogeneity is found in the decrease of the Shannon entropy with  $\sigma_0^2$ . The normalized mean Shannon entropy of  $H^+$  is consistently lower than that of  $H_2CO_3$ , again signifying that  $H^+$  is more associated with the preferential flow paths, thus exhibiting higher internal organization in the field.

The differences in the spatial distribution of reactants imply different traits also in the spatial distribution of dissolution-485 precipitation reactive events, associated with an abundance of  $H^+$  and  $H_2CO_3$ , respectively. These traits suggest that more dissolution events occur within the preferential flow paths due to their higher internal organization, while precipitation events occur more at the periphery. Different spatial patterns for dissolution and precipitation reactions in reactive flow in heterogeneous / fractured porous media have been reported both experimentally (Liu et al., 2005, Jones and Detwiler, 2016) and nu-490 merically (Edery et al., 2021, Kohlhaas et al., 2025, Kaufmann et al., 2016, Szawełło et al., 2024). For dissolution-dominated flows, as in our case, dissolution occurs mostly within the preferential flow paths, characterized by higher permeability and flow velocity and, thus, ample supply of reactive species. Higher flow velocity within the paths limits the residence time and, thus, the amount of precipitation within the path itself. Precipitation within the paths is also constrained by the more uniform transport there that inhibits mixing, thus allowing less chance to disrupt the local chemical equilibrium. However, at the periphery of the paths, less permeable as a rule, residence time increases as well as the chance to disrupt local chemical equilibrium, 495 thus leading to an increased precipitation there. Both these processes further enhance the preferential flow paths, making the paths themselves more permeable by virtue of excessive dissolution there, thus attracting even more reactant within the paths, along with making the periphery of the paths even less permeable via an increased precipitation there.

# 4.2 Non-equilibrium thermodynamic framework: analysis of the entropy generation sources

Having established that emergence and intensification of preferential flow paths in dissolution-dominated chemical weathering is a self-enhancing process that leads to an increased transport self-organization in the matrix, as a next step we quantify the dissipative dynamics of chemical weathering using concepts from Section 3.1. Neglecting temperature variations and the ensuing heat transfer effects in the porous matrix, we concentrate on the interrelated processes of *percolation*, *mixing* and *chemical reaction*. We analyze the entropy generation rates attributed to these processes to understand the evolution of our thermodynamic system due to internal interactions between them. The expressions for local entropy generation, corresponding to individual terms in (20), are integrated over the computational field to obtain the *total* entropy generation rates. We then normalize these rates by the quantity  $T_{ref} R \dot{N}_{tot}$  to account for deviations in the total flow rate through the field, obtained in SGSIM-generated hydraulic conductivity realizations for different  $\sigma_0^2$  values. Here,  $T_{ref}$  is the reference temperature taken to be 293.15K, R is the universal gas constant and  $\dot{N}_{tot}$  is the total molar flow rate into the field. The obtained dimensionless quantity represents the total entropy generation rate in the field per unit flow through the matrix (see Supplementary 3 for calculation details).

We first consider the **Reactive** entropy generation, calculated according to (S8)–(S9) in Supplementary 3. A characteristic value of 3e-4  $[{
m M}\,{
m sec}^{-1}]$  is used for the forward reaction rate  $r_{f,eq}$  of calcite dissolution at standard atmospheric conditions,

estimated based on data and procedure in Peng et al., 2015. Since the reaction in the field is reversible and occurs in both the dissolution and precipitation directions, some of the reactive events cancel each other out due to the immediate balancing with others in the opposite direction in the same cell during the time step, without eventually contributing to the alteration of the transport characteristics of the porous matrix. The reactive events that are not being canceled out immediately are the ones that affect the porosity and the hydraulic conductivity distributions in the field, leading to the intensification of preferential flow paths (Shavelzon and Edery, 2024). We are interested in determining that specific part of the reactive events and the ensuing *useful* Reaction entropy. We approach this task by calculating the total porosity increment in the field between each pair of consecutive time steps, while considering the absolute value of both the positive and negative porosity contributions. This is because both dissolution and precipitation reactive events contribute to the overall heterogeneity of the porous matrix, which is found to be in direct correlation with transport self-organization (Zehe et al., 2021, Shavelzon and Edery, 2024). Knowing the value of a porosity increment due to a single reactive event in a cell, defined in (9), we obtain the number of *useful* reactive events that occurred during the current time step, such that contributed to the alteration of transport characteristics of the porous matrix. Converting this number into the corresponding extent of reaction, using (S8)–(S9) in Supplementary 3 and normalizing as described above, the total normalized *useful* Reaction entropy generation rate in the field is obtained.

Figure 6a presents the total normalized Reactive entropy generation rate in the matrix  $\dot{S}_{qen,react}/(R\dot{N}_{tot})$  as a function of dimensionless time  $\tilde{t}$ , for varying  $\sigma_0^2$  values. Here, a localized area of first 10 columns of computational cells near the inlet was left out from the calculation to account for the massive dissolution there due to low-pH water injection. We observe that Reactive entropy generation increases monotonously until some value of  $\tilde{t}$  and then changes its behavior drastically, either reaching an approximately asymptotic value or decreasing, depending on the value of  $\sigma_0^2$ . This localized tipping point can be identified as a point at which enough reactants have traversed the field, thus establishing a complete flow path from inlet to outlet. Higher  $\sigma_0^2$  values lead to earlier flow path establishment. Before this point is reached, reactants travel towards the outlet, covering greater area and leading to an increase in the overall reaction intensity. After this point, preferential flow paths have been established and reactive events become more internally organized within these paths, leading to a substantial decrease in the overall reaction intensity in the matrix per unit time. This leads to a decline in the normalized Reaction entropy generation rate, directly correlating with  $\sigma_0^2$  as a measure of transport channelization in the field. While before the tipping point the entropy generation rate was higher for higher  $\sigma_0^2$  values, this tendency has been overturned afterwards. This can be explained by the fact that in the beginning reactants advanced faster within the preferential flow paths, which allowed them more opportunities for reaction as opposed to slower movement in the less heterogeneous fields. However, after the preferential flow paths have been fully established, in the more heterogeneous fields the reaction has been confined mostly to the close neighborhood of these paths, which reduced the opportunities for reaction, while for lower  $\sigma_0^2$  it proceeded to occur more homogeneously.

Figure 6b presents the normalized Reactive entropy generation rate  $\dot{S}_{gen,react}/(R\dot{N}_{tot})$  at  $\tilde{t}=2.0$  as a function of normalized distance from inlet  $\tilde{x}$ . The tendencies shown in Figure 6a are present also here, as the entropy generation is constantly higher for lower values of  $\sigma_0^2$ . The spatial distribution of Reactive entropy generation exhibits a decline with distance from in-

Figure 6. Entropy generation: (a) total normalized *Reactive* entropy generation rate over the field as a function of dimensionless time  $\tilde{t}$ , (b) normalized *Reactive* entropy generation at  $\tilde{t}=2.0$  as a function of normalized distance from inlet  $\tilde{x}$ , (c) total normalized *Percolative* entropy generation rate over the field as a function of dimensionless time  $\tilde{t}$ , (d) normalized *Percolative* entropy generation at  $\tilde{t}=2.0$  as a function of normalized distance from inlet  $\tilde{x}$ , (e) total normalized *Mixing* entropy generation rate over the field as a function of dimensionless time  $\tilde{t}$ , (f) normalized *Mixing* entropy generation at  $\tilde{t}=2.0$  as a function of normalized distance from inlet  $\tilde{x}$ .

let, hinting at an increase in transport channelization that inhibits the reaction, this channelization more pronounced for higher  $\sigma_0^2$  values. The Reactive entropy generation results presented in Figure 6a-b correlate well with Figure 5, drawing a clear correlation between the intensification of preferential flow paths and the ensuing inhibition of chemical reaction in the matrix.

Next, we analyze the **Percolative** entropy generation, representing the loss of hydraulic energy due to frictional dissipation in the matrix. Figure 6c shows the total normalized Percolative entropy generation rate over the field  $\dot{S}_{gen,perc}/(R\,\dot{N}_{tot})$  as a function of dimensionless time  $\tilde{t}$  for different  $\sigma_0^2$  values, calculated accordingly to (S4)–(S5) in Supplementary 3 and normalized by the quantity  $T_{ref}\,R\,\dot{N}_{tot}$ . We observe that the normalized Percolative entropy generation rate decreases with the dimensionless time  $\tilde{t}$ , since the preferential flow paths in the matrix become more conductive with time due to dominant dissolution that takes place there. This mechanism also allows to explain the decrease in the normalized Percolative entropy with the increase in  $\sigma_0^2$  value, as more heterogeneous matrix leads to more dominant preferential flow paths. This allows the flow to move more easily through the established flow paths, which are more conductive than the rest of the field, thus reducing the free energy losses associated with frictional dissipation per unit flow through the matrix.

Figure 6d presents the normalized Percolative entropy generation rate at  $\tilde{t}=2.0$  as a function of normalized distance from inlet  $\tilde{x}$ . As a rule, the Percolative entropy generation for lower  $\sigma_0^2$  values is higher for most of the field, corresponding to the tendencies in Figure 6c. Here, as opposed to the Reactive entropy generation, no clear spatial tendencies are observed, as the Percolative entropy generation rate remains approximately constant, except for local statistical fluctuations due to the heterogeneity of the porous medium (the fact that these fluctuations occur in similar regions for different  $\sigma_0$  values is not surprising, as the fields were created with identical seeds for SGSIM). This brings to mind intriguing associations with the Optimal channel network theory (OCN) and its application to drainage networks, as equal energy expenditure due to frictional dissipative process per unit channel area anywhere in the network is postulated by Rodriguez-Iturbe et al., 1992 as one of the three governing principles of optimal energy expenditure for a network. A disclaimer should be made that providing a full analogy to OCN is not intended in the current study, since to be able to do that we need to show that the principles of minimum energy expenditure in any link of the network and minimum total energy expenditure in the network as a whole are satisfied as well, which is well beyond the scope of this work. To summarize, the Percolative entropy generation results indicate lower free energy losses due to frictional dissipation through the matrix per unit flow as a result of the increasing transport channelization.

Finally, we consider the evolution of the **Mixing** entropy generation in the calcite matrix, representing the dissipation of chemical energy due to mixing of chemical constituents. This parameter was calculated using the *non-reactive* particle tracker numerical simulation (NRPT) in order to eliminate chemical reaction effects (see Supplementary 3 for details). A total of 1e5 particles representing the non-reactive tracer were injected in the snapshots of hydraulic conductivity field that represents the calcite matrix undergoing the reaction-transport interaction, taken at different times  $\tilde{t}$ . The flow in the field is governed by the boundary condition on the hydraulic head drop of  $\Delta h_{BC} = 100 [{\rm cm}]$ . This setting represents the physical scenario of mixing of two miscible species, such as water and non-reactive tracer. The obtained cumulative concentration distribution of the non-reactive tracer across the field was then used to calculate the Mixing entropy generation. Mixing entropy, obtained from NRPT simulations, can be considered an auxiliary parameter that quantifies dispersivity in the field, therefore representing the thermodynamic counterpart to the Shannon entropy of transport self-organization that was also calculated from the concentration distribution of reactants.

Figure 6e shows the total normalized Mixing entropy generation rate in the field,  $\dot{S}_{gen,mix}/(R\dot{N}_{tot})$ , as a function of dimensionless time  $\tilde{t}$  for different  $\sigma_0^2$  values, calculated accordingly to (S18)–(S19) in Supplementary 3 and normalized by the quantity  $T_{ref}R\dot{N}_{tot}$ . We observe that the normalized Mixing entropy increases with  $\sigma_0^2$ , signifying an increase in the gradients of chemical potential and concentration in the matrix. This can be attributed directly to the intensity of transport channelization in the field, as greater channelization, corresponding to higher  $\sigma_0^2$  values, attenuates the mixing processes in the field, causing the particles to move in a more organized way within the established preferential flow paths. This correlates well with the spatial tendencies for Reaction entropy, shown in Figure 6a.

Figure 6f presents the normalized Mixing entropy generation rate at  $\tilde{t}=2.0$  as a function of normalized distance from inlet  $\tilde{x}$ . The increase in the cumulative Mixing entropy generation downstream indicates an increase in the gradients of chemical potential and concentration with distance from inlet, signifying an intensifying channelization in the downstream direction. The observed tendencies for the Mixing entropy generation support our hypothesis regarding the Reaction entropy, showcasing that intensification of transport channelization in preferential flow paths inhibits the mixing of the reactants, thus leading to a decline in the overall reaction rate.

A general remark regarding the tendencies, exhibited by the Mixing entropy generation. As a rule, the expressions for entropy generation involve the product of a thermodynamic potential gradient (such as hydrodynamic pressure) and the corresponding thermodynamic flux (such as Darcy flux) - see Section 3.1 and Supplementary 2. Consider as an example the Percolative entropy: in case of injection of low-pH water into calcite matrix, dissolution of the matrix will be obtained, corresponding to an increase in the overall hydraulic conductivity. This will result in a decrease in energy dissipation per unit flow rate due to viscous friction, as smaller pressure gradient will be required to drive the same fluid flux through matrix. In case of boundary condition on the thermodynamic potential (constant pressure gradient over the matrix, as in our model), the total flow rate through matrix will increase and the resulting Percolative entropy generation should be normalized by this total flow rate, otherwise an increase in the entropy generation rate will be obtained. Regarding the mixing of chemical species during the intensification of preferential flow paths, a similar effect is obtained: the ensuing transport channelization, leading to a decrease in mixing in the matrix, is expressed via an increase in concentration gradients. By looking at (S18), it is clear that the Mixing entropy generation will increase in this case, as observed in Figures 6e–f for an increase in  $\sigma_0^2$ , as well with an increasing distance from inlet.

The Mixing entropy obviously has a certain correlation with the Shannon entropy of transport self-organization, as the physical interpretation of the two can be very close. An important distinction between the two quantities, as noted in Ben-Naim, 2017, lies in the broadness of applicability: while the Shannon entropy can defined on any statistical distribution, making it a very general concept, the Mixing entropy (or any other source of entropy generation) is applicable to a specific set of parameters defined for the specific physical process or system. It should be noted, however, that the general discussion regarding comparison and possible interpretations of these two quantities is far beyond the scope of the current study. For an in-depth discussion,

see Ben-Naim, 2017 and the series of articles on this topic by Arieh Ben-Naim.

To summarize, our results show a clear correlation between the emergence and intensification of preferential flow paths and the accompanying dissipative dynamics, where the evolution of the emerging paths leads to a decrease in the free-energy dissipation per unit flow rate due to flow percolation, mixing of chemical constituents and reaction. Based on the tendencies exhibited by the Reactive and Percolative entropy generation rates, we may conclude that the emergence of preferential flow paths due to chemical weathering in geophysical systems represents an energetically-preferred state of the system that can be considered a manifestation of the minimum energy dissipation principle (Kondepudi and Prigogine, 1998). This allows interpreting the process of chemical weathering as an evolution of a non-equilibrium system towards a stationary state under the applied constraint of an influx of reactive fluid into the medium. We are able to speculate about the nature of this non-equilibrium stationary state that corresponds to a complete channelization of the medium, thus minimizing the mixing of reactive species, inhibiting the ensuing chemical reaction, and providing conductive paths for the flow. Our combined approach to characterize the complex reaction-transport interaction during geochemical weathering within the non-equilibrium thermodynamic framework constitutes the main point of our manuscript. The obtained correlation may serve as a general concept in geophysical dating

A remark regarding future work directions. The above results show that reaction-transport interaction in heterogeneous porous medium is characterized by the formation of highly intense preferential flow paths that play a dominant role in both the reactive and transport processes in the medium. Thus, it makes sense to concentrate on these "hot spots" in the analysis, as their influence is much more dominant than that of the rest of the field. This brings to mind the global-local nesting, or nested grid approach, where an approximate global-scale solution is obtained on a coarse mesh, while an accurate local-scale solution within an element of interest is computed on a fine mesh using boundary conditions obtained from the coarse solution. This reduces significantly the computational costs, while allowing to account for the complex multi-scale nature of the porous media (Gautier et al., 1999). For more application of global-local nesting approach for solution of multi-scale problems in Earth sciences, see Shavelzon and Givoli, 2009, where this approach was applied in the context of atmospheric weather prediction models, and Van Sy et al., 2016 in the field of oceanic circulation models. Investigation of reactive flow in heterogeneous porous media, dominated by the preferential flow path phenomenon, by employing the nested grid approach is a subject of future work.

#### 4.3 Implications for subsurface carbon sequestration by mineralization

The relevance of our study can be exemplified on a case of subsurface carbon sequestration by mineralization that involves injection of either dissolved or supercritical  $CO_2$  into reactive rocks, particularly mafic, ultramafic or carbonate, leading to  $CO_2$  mineralization and fixing carbon in the subsurface with little risk of escape to the atmosphere (Snaebjornsdottir et al., 2020, Ruprecht and Falta, 2012). For the supercritical  $CO_2$  injection,  $CO_2$  solubility in brine increases with pressure as the solute advances deeper into subsurface, thus leading to acidification of the brine layer in contact with the supercritical  $CO_2$  (Pradhan et al., 2025). This enriched layer, having a typical pH of 3–5, is denser than the surrounding resident brine, thus

creating a negative buoyancy. This facilitates the transport of acidic brine downwards in the porous matrix (Ahmad et al., 2016). This acidic solution promotes the dissolution of silicate minerals, facilitating the subsequent  $CO_2$  mineralization by (a) consuming the hydrogen ions, which neutralizes the acidic solution and facilitates precipitation of carbonate minerals, and (b) providing cations that react with the dissolved  $CO_2$  to form stable carbonate minerals (Snaebjornsdottir et al., 2020). Thus, a complex reaction-transport interaction ensues following  $CO_2$  injection into subsurface, where the rate of  $CO_2$  mineralization is strongly affected by the preferential flow paths, either existing in the subsurface or emerged/altered due to dissolution of silicate minerals. Although  $CO_2$  mineralization in mafic rocks involves dissolved cations different from  $Ca^{2+}$  as in our study  $(Mg^{2+}, Fe^{2+}, \text{ etc.})$ , nevertheless the mechanism follows the same traits, thus allowing us to speculate on this subject based on our study.

The existence of dominant preferential flow paths and their intensification due to reaction-transport interaction can facilitate  $CO_2$  injection and may be favorable for carbon sequestration as it allows decreasing the hydraulic power per unit flow rate required for  $CO_2$  injection (Oldenburg, 2001). This is confirmed by Figure 6c in the manuscript that shows a decrease in the normalized Percolative entropy with the intensification of preferential flow paths due to either an increase in the field heterogeneity  $\sigma_0^2$  or the computational time  $\tilde{t}$ , which signifies a decrease in the free-energy dissipation due to flow percolation. On the other hand, for  $CO_2$  mineralization, the existence of dominant preferential flow paths may inhibit the overall extent of reaction per unit flow rate due to channelization of reactive species that limits their exposure to calcite (Harrison et al., 2016). This is confirmed by Figure 6a that shows a decrease in the normalized Reactive entropy with the intensification of preferential flow paths, signifying a decrease in the overall reaction rate. To summarize, for a given pressure head, an injection into a more heterogeneous matrix will result in a higher injection rate, while a more homogeneous domain will yield a higher mineralization rate, thus exemplifying the resulting trade-off in the injection strategy.

# 5 Conclusions

Our computational study analyzes the evolution of reaction-transport interaction on a Darcy scale, representative of chemical weathering of calcite porous matrix, in artificially generated fields of varying degrees of heterogeneity employing the non-equilibrium thermodynamic framework. In the simulated scenario, the reaction is caused by the injection of low-pH water in a calcite matrix, initially in chemical equilibrium with the resident fluid, leading to a reversible dissolution-precipitation reaction with the matrix. We identify the entropy generation sources, attributed to the dissipative processes inherent to this physical scenario and investigate the correlation between the emergence and intensification of preferential flow paths, characterized using Shannon entropy, and the accompanying dissipative dynamics due to flow percolation, mixing of chemical constituents and reaction. Our work leads to the following key conclusions:

Different traits in the spatial distribution of dissolution-precipitation reactive events are established, as dissolution events
occur more within the preferential flow paths that act as main channels for supply of reactants, while precipitation
events occur more at their periphery. This leads to intensification of preferential flow paths, as the paths themselves are

made more conductive by virtue of excessive dissolution, while their periphery becomes less permeable via an increased precipitation.

- By analyzing the entropy generation rates attributed to energy dissipation mechanisms pertinent to the physical scenario of chemical weathering, we obtain a clear correlation between the emergence and intensification of preferential flow paths and the accompanying dissipative dynamics, where the evolution of the emerging paths leads to a decrease in the free-energy dissipation rate due to flow percolation, mixing of chemical constituents and reaction. Based on the tendencies exhibited by the Reactive and Percolative entropy generation rates, we may conclude that the emergence of preferential flow paths due to chemical weathering in geophysical systems represents an energetically-preferred state of the system that can be considered a manifestation of the minimum energy dissipation principle (Kondepudi and Prigogine, 1998). This allows interpreting the process of chemical weathering as an evolution of a non-equilibrium system towards a stationary state under the applied constraint of an influx of reactive fluid into the medium. We are able to speculate about the nature of this non-equilibrium stationary state that corresponds to a complete channelization of the medium, thus minimizing the mixing of reactive species, inhibiting the ensuing chemical reaction, and providing conductive paths for the flow, which are preferable energetically.
- The relevance of our study was exemplified on a case of subsurface carbon sequestration by mineralization that involves injection of dissolved or supercritical CO<sub>2</sub> into reactive rocks, leading to CO<sub>2</sub> mineralization. Based on the established results, we conclude that, for a given pressure head, an injection into a more heterogeneous matrix will result in a higher injection rate, while a more homogeneous domain will yield a higher mineralization rate, thus exemplifying the resulting trade-off in the injection strategy.
- Our combined approach to characterize the complex reaction-transport interaction during chemical weathering, which combines seemingly distant concepts from non-equilibrium thermodynamics and information theory, constitutes the main point of this manuscript. This approach allows for a deep physical insight into chemical weathering, allowing to correlate between the evolution of preferential flow paths and the dissipative dynamics in the system. The obtained results may prove useful in determining optimal conditions for pressurized fluid injection into subsurface, as well as in geophysical dating.
- Future work will include application of the nested grid approach to simulation of reaction-transport interaction in a heterogeneous porous medium, dominated by preferential flow paths. This will allow reducing significantly the computational costs, while properly accounting for the complex multi-scale nature of transport in porous media.

Code and data availability. Software and data archiving is currently underway. For review purposes, a copy of software and data has been uploaded to a dedicated GitHub repository under the following address: https://github.com/PMVlab/PMVlab-Evolution-of-preferential-flowpaths-non-equilibrium-thermo-PhD-paper2-SOFTWARE.

Author contributions. E.S. - Conceptualization, Data curation, Formal analysis, Investigation, Methodology, Software, Validation, Visualization, Writing - original draft, Writing - review & editing. E.Z. - Conceptualization, Methodology, Supervision, Writing - review & editing. Y.E. - Conceptualization, Funding acquisition, Methodology, Project administration, Software, Resources, Supervision, Writing - review & editing.

Competing interests. At least one of the (co-)authors is a member of the editorial board of Hydrology and Earth System Sciences.

*Acknowledgements.* YE and ES acknowledge the support of the Israel Science Foundation (grant no. 3774/24). E.Z. gratefully acknowledges support through the ViTamins project, funded by the Volkswagen Foundation (Grant No. AZ 9B192).

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
