# Peer review of "Linking chemical weathering, evolution of preferential flow paths and transport self-organization in porous media using non-equilibrium thermodynamics"

_EGUsphere, 2025_

## Referee Comment (RC2)

The manuscript by Shavelzon et al. aims to establish a novel connection between chemical weathering, transport self-organization, and non-equilibrium thermodynamics by using a particle-tracking reactive transport model in 2D porous media with varying heterogeneity. This topic would likely be of interest to researchers seeking better frameworks to describe the formation of preferential pathways in the subsurface driven by water-rock interactions. However, the current version of the manuscript is difficult to read and interpret, due to excessive technical density, insufficient narrative structure, and limited discussion of the model assumptions and broader scientific implications. I think major revisions are needed before the manuscript can be considered for publication.

1. Excessive length, technical complexity and abstract terminology. The paper dives into lengthy derivations and modeling details early on (especially Sections 2-3), but without sufficient explanations for why this framework matters or what the reader should expect to learn. This makes it difficult to parse how the pieces fit together—especially for readers unfamiliar with entropy generation in porous media. A conceptual figure or roadmap either at the end of the Introduction or at the beginning of the Methods would help guide the readers.

2. Unclear assumptions in reactive transport setup and their implications for the conclusions. The authors employ a simplified reactive system consisting of only two solute species ($H^+$ and $H_2CO_3$) and a single dissolution–precipitation reaction involving calcite. Important geochemical complexities, such as full aqueous speciation (e.g., bicarbonate, carbonate), are omitted. I think it is necessary to clarify the rationale for these assumptions and discuss their implications for the applicability of results to real-world weathering environments.

3. Conclusions are too technically narrow. It would better for the authors to frame their findings in the broader implications for real-world geochemical systems, such as rock weathering in karst systems or channelization in $CO_2$-enhanced weathering. Discussing such applications would help connect the modeling results to field-scale processes and enhance the overall significance of the study.

Minor comments:
- Abstract, Line 2: "that" -> "which"; add "the" before "emergence"
- Introduction, Line 17: It is not appropriate to refer to chemical weathering as a geophysical phenomenon. Consider revising to "geochemical process"
- Line 663: typo "expend Figure 8biture for the network"

---

## Author Comment (AC2)

**Linking chemical weathering, evolution of preferential flow paths and transport self-organization in porous media using non-equilibrium thermodynamics**

Evgeny Shavelzon[1], Erwin Zehe[2], and Yaniv Edery[1]

[1]Faculty of Civil and Environmental Engineering, Technion - Israeli Institute of Technology, Haifa, Israel
[2]Karlsruhe Institute of Technology (KIT), Institute of Water and River Basin Management, Karlsruhe, Germany

**Correspondence:** Yaniv Edery(yanivedery@technion.ac.il)

**Replies to Reviewer 1**

We appreciate the time and effort invested by the respectable Reviewer in our manuscript and are grateful for the opportunity to resubmit a revised draft for your consideration. We address all Reviewer's comments in the following itemized list, arranged such that the answer to each review item is located directly below the item. Please note that Reviewer's comments and our responses are given in black and blue, respectively. Sincerely, Evgeny Shavelzon, Erwin Zehe and Yaniv Edery.

**Reviewer 1**

– This manuscript is about the dissolution of calcite in porous media and the resulting self-organization of preferential flow paths. The results are based on numerical simulations and are analyzed with the help of thermodynamic variables. Although the basis of the model seems to be similar to what has been assumed in models of karst evolution for decades, I found this study very interesting. The are numerous points to think about. On the other hand, I find it a bit too difficult. Each step is not really difficult in itself, but in sum it is very challenging not to get lost. When reading it, I had to go back quite often to find out whether something is indeed not clear or whether I just missed it. So, I am a bit afraid that scientists who did not have to hold on to write a review will likely give up.

We thank the Reviewer for their careful review and positive comments that were extremely helpful in clarifying and improving the readability of the manuscript. Indeed, as the reviewer has pointed out, the main point of the study is the analysis of the process of geochemical weathering within thermodynamic framework and establishing the correlation between chemical weathering and the emergence of preferential flowpaths in the system, rather than the modeling of the geochemical process itself, which has been done previously. We are thankful to the reviewer for indicating the need to improve the readability of the manuscript, clarifying it and making it more concise. We fully acknowledge this need and

are committed to amending the manuscript accordingly. Please find attached the summary of our suggestions regarding improving the readability towards the end of our reply.

– So let me just make some suggestions, hoping that they may be helpful for clarifying some aspects. Quite in the beginning, I wondered about the porosity and permeability (Eqs. 7 and 8). If I got it correctly, you assume constant initial permeability and introduce spatial variability be assuming a variable factor in the Kozeny-Carman relation. This looks unconventional. The conventional approach would be the same factor, but variable initial porosity. I guess that you used this approach in order to define the pore volume time. However, I think that it also has an effect on how the development of the permeability. In your approach, a given change in porosity (from the constant initial value) introduces the same relative change in permeability. In the more conventional approach, the relative change would be smaller for cells that already have a high permeability. So I would guess that the formation of preferential flow paths in stronger in your approach than in the conventional version. This makes we wonder whether it has an effect on the results. Maybe you can clarify this aspect.

We thank the reviewer for addressing this important point. The constant initial porosity assumption was introduced in order to facilitate the chemical aspects of our reactive transport model, namely the calculation of the molar amount of $H^+$ to give pH of 3.5 in a computational cell. This allows to estimate the current pH in the cell and, thus, the direction and the extent of reaction there (see Section 2.3 in Shavelzon and Edery (2024)). This assumption was deemed acceptable, as the purpose of our simplified chemical reaction setup is to capture the qualitative dynamics of the complex process of geochemical weathering, thus making it possible to simulate and analyze the complex reversible behavior using a relatively unsophisticated model.

In our model, the change in porosity of a specific cell in the computational field is determined by the amount of reaction that has occurred in a cell during the current time step, as dissolution/precipitation of calcite increases/reduces the void volume of the cell (as expressed by Eq.(7)). Given the initial and the updated values of porosity in the cell, as well as the initial permeability value, we are then able to calculate the updated permeability in the cell using the Kozeny-Carman relation as shown in Eq. (8). Notice, that eq. (8) includes both initial and updated porosity values in a nonlinear relation that cannot be expressed as a function of only a change in porosity. Therefore, the relative change in permeability does depend on the initial porosity value. To demonstrate this, we attach below the plot of the Relative change in permeability as a function of porosity $K_{k+1}/K_k$, as given by the expression in Eq. (8) as a function of initial porosity $\theta_k$, for the change in porosity $\Delta\theta = 0.05$ - see Figure 1. We observe that for larger $\theta_k$ values, that correspond to higher permeability, the relative change in permeability is smaller, just as is physically expected. In our model, we make a simplifying assumption of constant initial porosity value of 0.43 for all field realizations, regardless of their heterogeneity degree, while the permeability is distributed randomly based on prescribed values of distribution variance and correlation length as explained in Section 2. While this assumption cannot be seen as rigorous, we argue that it doesn't impact significantly

[Figure]

**Figure 1.** Relative change in permeability as a function of initial porosity.

the simulation results, as well as the main conclusions made, since the main point of the manuscript is to capture the essential behavior of the complex coupled process of geochemical weathering, simple enough to not make the reactive model the main issue of the manuscript, but sufficient to be able to draw conclusions about the implications of the ensuing reaction-transport interaction.

We propose to add the following paragraph to the manuscript to clarify this point:

*The constant initial porosity assumption, introduced in order to facilitate the chemical aspects of our reactive transport model, was deemed acceptable, as the purpose of our simplified chemical reaction setup is to capture the qualitative dynamics of the complex process of geochemical weathering, thus making it possible to simulate and analyze the complex reversible behavior using a relatively unsophisticated model.*

– My main concern is, however, that the entropies are somewhat abstract properties.

We are sincerely thankful to the Reviewer for pointing out the need to clarify the meaning of both the Shannon and the physical entropies that are employed extensively in the manuscript. In the manuscript, we are looking for the correlation between the emergence and intensification of preferential flowpaths and the dissipation of the useful energy of the

system due to different processes, characteristic for chemical weathering, such as chemical reaction, mixing of chemical species and percolation (frictional dissipation due to fluid viscosity). To quantify the intensity of the preferential flowpaths phenomenon (how much is the transport concentrated, or self-organized, within these paths, as opposed to a homogeneous distribution throughout the field), we employ the Shannon entropy, or information entropy – a quantity borrowed from information theory and used, among others, to quantify self-organization in physical systems (see the original paper on this subject by Shannon (1940), as well as Zehe et al (2021) and Shavelzon and Edery (2024) for application to transport self-organization in porous media). On the other hand, to quantify the useful energy dissipation during chemical weathering, we employ the physical entropy, a quantity that, within the framework of nonequilibrium thermodynamics, allows to quantify energy dissipation due to chemical reaction, mixing of chemical species and percolation – see, for instance, Kondepudi and Prigogine (1998). Thus, by employing quantities from seemingly unrelated disciplines, we attempt to correlate between these two parameters in order to obtain a deeper understanding of the phenomenon of preferential flowpaths. These quantities, however, are far from being abstract, but have a precise physical meaning. The main conclusion we arrive at after analyzing both the Shannon entropy of transport self-organization and the physical entropy sources is that a clear correlation is established between the emergence and intensification of preferential flowpaths and the useful energy dissipation due to reaction, mixing and percolation in geochemical systems that involve chemical weathering. The emergence of preferential flowpaths allows the system to reduce this useful energy dissipation rate and, thus, represents an energetically preferred state of the system.

To improve the clarity of presentation of a non-equilibrium thermodynamic framework in the manuscript, we propose adding a short chapter that explains in simple terms the topic of dissipative processes in non-equilibrium thermodynamics to the Supplementary. See **Appendix A: Non-equilibrium thermodynamics in a nutshell** in the current document.

We also propose adding a short section that briefly explains the concept of the Shannon entropy and its use for quantifying self-organization in physical systems, as well as its relevance to our physical scenario, along the lines of Supplementary 2 in Shavelzon and Edery (2024) (currently, the manuscript draft references this publication, where this subject is treated extensively). Depending on the reviewer's opinion, this section may be either rewritten and added as a Supplementary material to the current manuscript or, as an alternative option, references to Supplementary 2 in Shavelzon and Edery (2024) may be given in the appropriate places in the manuscript. In that way, both the Shannon and the physical entropy, as well as their relevance in our manuscript, would be explained in the Methodology section before any use of them is implemented in the Results. Full text of Supplementary 2 in Shavelzon and Edery (2024) is given in the **Appendix B: Employing Shannon entropy to quantify transport self-organization** in the current document. We also propose to summarize this discussion in manuscript by add the following paragraph:

*We are looking for the correlation between the emergence and intensification of preferential flowpaths and the dissipation of the useful energy of the system due to different processes, characteristic for chemical weathering, such as*

*chemical reaction, mixing of chemical species and percolation (frictional dissipation due to fluid viscosity). To quantify the intensity of the preferential flowpaths phenomenon, we employ the Shannon entropy, or information entropy – a quantity borrowed from information theory and used, among others, to quantify self-organization in physical systems. On the other hand, to quantify the useful energy dissipation during chemical weathering, we employ the physical entropy, a quantity that, within the framework of nonequilibrium thermodynamics, allows to quantify energy dissipation due to chemical reaction, mixing of chemical species and percolation. Thus, by employing quantities from seemingly unrelated disciplines, we attempt to correlate between these two parameters in order to obtain a deeper understanding of the phenomenon of preferential flowpaths.*

– It all makes sense, but I am still left with the feeling that I do not understand fully what happens in the system. One point is why H+ is associated with the preferential flow paths more closely than H2CO3.

We are sincerely thankful to the Reviewer for pointing out this subject, that has not received a sufficient explanation in the manuscript. Different spatial patterns for dissolution and precipitation reactions in reactive flow in heterogeneous / fractured porous media have been reported both experimentally (Liu et al (2005), Jones and Detwiller (2016)) and numerically (Edery et al (2021), Kohlhaas et al (2025), Kaufmann et al (2016), Szawello et al (2024)). For dissolution-dominated flows, such as in our case, dissolution occurs mostly within the preferential flowpaths, characterized by higher permeability and flow velocity and, thus, ample supply of reactant (these paths act as main channels for reactant supply). Higher flow velocity within the paths limits the residence time and, thus, the amount of precipitation within the path itself (which is also limited by the more uniform transport within the path that limits mixing, therefore allowing for less chance to disrupt the local chemical equilibrium). However, at the periphery of the paths, which is less permeable as a rule, residence time increases as well as the chance to disrupt local chemical equilibrium, thus leading to increased precipitation there. Both these processes further enhance the preferential flowpaths by making the paths themselves more permeable by virtue of excessive dissolution there, thus attracting even more reactant within the paths, along with making the periphery of the paths even less permeable via an increased precipitation there. This preferential flowpaths enhancement leads to transport self-organization, or channelization, as attested by the decrease in Shannon entropy over time (see Figure 4).

In our manuscript, we simulate the dissolution-dominated reactive flow scenario by injecting low-pH water into the calcite porous matrix. In our chemical reaction model, dissolution of calcite is associated with an abundance of hydrogen ion H+, therefore it is reasonable that it will self-organize within the preferential flow paths, dominated by the dissolution reaction. On the other hand, H2CO3, responsible for precipitation in our model, is expected to be less associated with the preferential flowpaths but rather to exhibit a more homogeneous distribution due to increased precipitation in the vicinity of these paths. Following the respectable Reviewer's comment, we propose to amend the manuscript by adding this discussion in Results/Conclusions.

140 – In this context, I would find it very helpful to separate the propagation of the reaction front from the changes in the flow pattern. So considering the dynamics of the reaction without taking into account the effect on the permeability. Then it would be easier to understand what exactly the effect of the changes in permeability is. Maybe the dynamics of the reaction is much faster and it is already clear, but I am not sure.

145 We sincerely thank the reviewer for suggesting ways to clarify the manuscript. However, we would like to draw the reviewer's attention to the fact that the main point of the manuscript is is the specific interaction between the reactive and transport processes in the calcite porous matrix that leads to the emergence of preferential flowpaths in the matrix. Thus, we will not gain any additional insight by examining the reaction separately from the changes in permeability, as within the particle tracker simulation we cannot account for the reaction without altering the permeability in response

150 to it, since the dissolution-precipitation reaction alters the transport properties of the porous matrix. On the contrary, we aim at showing how the reaction-transport interaction affects both of these processes: the dissolution of calcite, that occurs in the preferential flow paths, intensifies these paths by making them more conductive (as shown by the decrease in Percolative entropy which signifies the decrease in frictional dissipation as the fluid makes its way through the porous matrix) and, therefore, increases the channelization of transport in these paths (as shown by the decreasing Shannon

155 entropy of transport self-organization). On the other hand, this channelization of transport inhibits mixing of chemical species, therefore reducing the overall reaction rate (as shown by Mixing, Reactive entropies). We are committed to make necessary amendments to the manuscript to clarify this aspect.

– Perhaps it would also be helpful to complement the 2D pictures of the concentrations in Fig. 3 by longitudinal profiles (integrated along the y-axis) in order to see whether the species are just concentrated more or less along the preferential

160 flow paths or also follow strongly different distributions along the x-axis.

We thank again the reviewer for suggesting ways to further clarify the manuscript. The concentration of species along the preferential flowpaths, or, as we refer it in the paper, the intensity of the preferential flowpaths phenomenon (meaning how much is the spatial distribution of the species concentrated, or self-organized, within these paths, as opposed to a homogeneous distribution throughout the field), is quantified in the manuscript using the Shannon entropy (see explana-

165 tion on this quantity in the text above, as well as in the reply to the last item in the document). These trends are shown in Figure 4: the frames (a)-(c) show the Shannon entropy of spatial distributions of H+, H2CO3 and the total population as a function of distance from inlet at a specific time, while the frames (d)-(f) show the mean value of Shannon entropy of spatial distributions of H+, H2CO3 and the total population over the field as a function of time. In general, the decrease in Shannon entropy of some distribution means an increase in its spatial organization (Zehe et al (2021), Shavelzon and

170 Edery (2024)). The decrease in Shannon entropy with distance from inlet, as well as with an increase in variance of spatial permeability distribution $\sigma_0^2$, shows an increase in self-organization of species, meaning that their spatial distribution becomes less homogeneous, and, thus, more of the species find themselves within the preferential flowpaths. The

decrease in Shannon entropy with time also shows an increase in self-organization of species as the time passes, as a

175    result of reaction-transport interaction. We are committed to make necessary amendments to the manuscript to clarify this aspect.

– In summary: really interesting stuff, nothing wrong as far as I can see, but the paper would benefit from a more basic and clearer explanation of the model's behavior.

180    We thank the reviewer for expressing interest in the results presented in the manuscript, as well as for indicating the need to improve the readability of the manuscript, clarifying it and making it more concise. We fully acknowledge this need and are committed to amending the manuscript accordingly. Please find attached the summary of our suggestions regarding improving the overall clarity and readability of the manuscript, organized by chapter.

  (a) **Chapter 2 (Methodology)**:

185      • Adding a conceptual flowchart at the beginning of Methodology chapter - see Figure 2 in the current document. This flowchart details the various stages of the employed methodology, from creating a realization of hydraulic

[Figure]

**Figure 2.** Roadmap chart for the research procedure.

conductivity field to numerical simulation to calculating the Shannon entropy and the physical entropy sources employing data obtained from simulation. The flowchart also presents graphically the research question of

the manuscript, that concerns the correlation between the emergence of preferential flow paths (transport self-organization quantified using Shannon entropy) and the free-energy dissipation due to flow percolation, mixing and reaction (physical entropy sources). Each graphical block (shown by different colors in the flowchart) references the specific section where it is explained in details (section numbers have been changed to reflect the suggested manuscript amendments).

- Rewriting and expanding Section 2.2 (Chemical reaction model) to allow for a clearer description of the chemical model and its underlying assumptions. Also, we suggest minor corrections/additions to Chapter 2 as a whole to improve the overall presentation of methodology. See our suggestion for the updated Chapter 2 in **Appendix C: 2. Methodology** in the current document.

(b) **Chapter 3 (Nonequilibrium thermodynamic framework):**

- Shortening by moving Section 3.2 (Calculation of entropy generation terms) to Supplements, as it basically contains technical details that are not critical for understanding of the manuscript.

- Adding a short chapter that explains in simple terms the topic of dissipative processes in non-equilibrium thermodynamics to the Supplementary. See **Appendix A: Non-equilibrium thermodynamics in a nutshell** in the current document.

[Figure]

**Figure 3.** Sources of inflow and dissipation of free energy in subsurface reactive flow.

- Adding a short section that briefly explains the concept of the Shannon entropy and its use for quantifying self-organization in physical systems, as well as its relevance to our physical scenario, along the lines of Supplementary 2 in Shavelzon and Edery (2024) (currently, the manuscript draft references this publication, where this subject is treated extensively). Depending on the reviewer's opinion, this section may be either rewritten and added as a Supplementary material to the current manuscript or, as an alternative option, references to Supplementary 2 in Shavelzon and Edery (2024) may be given in the appropriate places in the manuscript. In that way, both the Shannon and the physical entropy, as well as their relevance in our manuscript, would be explained in the Methodology section before any use of them is implemented in the Results. Full text of Supplementary 2 in Shavelzon and Edery (2024) is given in the **Appendix B: Employing Shannon entropy to quantify transport self-organization** in the current document.

- Replacing Figure 1 in the manuscript with an improved version that explains more clearly the sources of free-energy inlfow into the system and its dissipation (see Figure 3 in the current document) Specifically, this figure visualizes the main result of Section 3.1: entropy generation, which is directly related to dissipation of the free-energy of the system, occurs in dissipative processes where the gradient of a thermodynamic potential drives the thermodynamic flux. Thus, the gradient of hydraulic head drives the mass flux, the gradient of the chemical potential drives the molar fluxes of the chemical species and the partial molar Gibbs energy drives the extent of reaction.

(c) **Section 4 (Results)**:

- Shortening by (a) moving Section 4.2 (Statistical analysis of the transport properties of the porous medium) to Supplements, as it contains results that are tangential to the main message of the manuscript, and (b) removing the discussion of the absolute Reactive entropy altogether and focusing on discussing the useful Reaction entropy, as this quantity can be measured experimentally.

- Adding a concluding paragraph allowing to frame our findings in the broader implications for real-world geo-chemical systems on an example of CO2 sequestration by mineralization, as a highly relevant example of an interaction between transport and dissolution-precipitation reaction. We propose to add the following to Results and Discussion:

*The relevance of our study can be exemplified on a case of subsurface carbon sequestration by mineralization, that involves injection of either dissolved or supercritical CO2 into reactive rocks, particularly mafic, ultramafic or carbonate, leading to CO2 mineralization and fixing carbon in the subsurface with little risk of escape to the atmosphere (Snaebjornsdottir et al(2020), Ruprecht and Falta (2012)). For the supercritical CO2 injection, CO2 solubility in brine increases with pressure as the solute advances deeper into subsurface, thus leading to acidification of the brine layer in contact with the supercritical CO2 (Pradhan et al (2025)). This enriched layer, having a typical pH of 3–5, is denser than the surrounding resident brine, thus creating a negative buoyancy. This facilitates the transport of acidic brine downwards in the porous matrix (Ahmad et al (2016)). This acidic solution promotes the dissolution of silicate minerals, facilitating the subsequent*

*CO2 mineralization by (a) consuming the hydrogen ions, which neutralizes the acidic solution and facilitates precipitation of carbonate minerals, and (b) providing cations that react with the dissolved CO2 to form stable carbonate minerals (Snaebjornsdottir et al(2020)).*

*Thus, a complex reaction-transport interaction ensues following CO2 injection into subsurface, where the rate of CO2 mineralization is strongly affected by the preferential flowpaths, either existing in the subsurface or emerged/altered due to dissolution of silicate minerals. Although CO2 mineralization in mafic rocks involves dissolved cations different from Ca(2+) as in our study (Mg(2+), Fe(2+), etc.), nevertheless the mechanism follows the same traits, thus allowing us to speculate on this subject based on our study.*

*The existence of dominant preferential flowpaths and their intensification due to reaction-transport interaction can facilitate CO2 injection and may be favorable for carbon sequestration as it allows decreasing the hydraulic power per unit flow rate required for CO2 injection (Oldenburg et al 2001). This is confirmed by Figure 8 in the manuscript that shows a decrease in the normalized Percolative entropy with the intensification of preferential flowpaths due to either an increase in field heterogeneity $\sigma_0^2$ or the reaction-transport interaction, which signifies a decrease in the free-energy dissipation due to flow percolation. On the other hand, for CO2 mineralization, the existence of dominant preferential flowpaths may inhibit the overall extent of reaction per unit flow rate due to channelization of reactive species which limits their exposure to the calcite matrix (Harrison et al (2015)). This is confirmed by Figure 6 in the manuscript that shows a decrease in normalized Reactive entropy with the intensification of preferential flowpaths, signifying a decrease in the overall reaction rate.*

*To conclude, our analysis implies that, for a given pressure head, a more homogeneous porous matrix will result in less pronounced preferential flowpaths, along with higher hydraulic energy dissipation and mineralization rates. On the other hand, for a highly heterogeneous matrix dominant preferential flowpaths will be obtained, along with lower hydraulic energy dissipation and mineralization rates. Considering these aspects for carbon sequestration where acidified brine leads to carbon mineralization, we conclude that, for a given pressure head, an injection into a more heterogeneous matrix will result in a higher injection rate, while a more homogeneous domain will yield a higher mineralization rate, thus exemplifying the resulting trade-off in the injection strategy.*

(d) **General suggestions:**

- Simplifying, where possible, the abstract terminology throughout the manuscript, such as the barycentric frame of reference (l.215), differential operators (l.240), Langevin equation paraphernalia (l.126-127), as related to purely technical aspects of our study (See Appendix C: Methodology in the current document).

- Improving and sharpening the narrative throughout the draft by amending the manuscript, including Section 1 (Introduction), in accordance with the updated Abstract (see below). The Abstract was updated to better emphasize the presented scenario of geochemical weathering and the implications of our study for this scenario:

**Abstract.** *Chemical weathering of soil and rock is a complex geophysical process during which the reaction and transport processes in the porous medium interact, causing erosion of the medium. This process is ubiquitous in geophysical systems and can be encountered, among others, in formation of karst systems, subsurface carbon sequestration and surface weathering of river beds. A common outcome of chemical weathering is the emergence and intensification of preferential flowpaths, where the weathering alters the transport properties of the rock, thus creating coupling between transport and reaction. While numerous approaches have been undertaken to simulate this complex interaction, still a need exists for a unified framework able to correlate the emergence of preferential flowpaths with the associated dissipative dynamics. As a case study, we consider subsurface chemical weathering of calcite porous rock undergoing reversible dissolution-precipitation reaction and apply non-equilibrium thermodynamic framework to analyze the ensuing reaction-transport interaction in this geophysical scenario. We identify the entropy generation sources, attributed to the dissipative processes inherent to this physical scenario and show a clear correlation between the emergence and intensification of preferential flowpaths and the accompanying dissipative dynamics, where the evolution of the emerging paths leads to a decrease in the free-energy dissipation rate due to flow percolation, mixing of chemical constituents and reaction. This indicates that the emergence of preferential flowpaths due to chemical weathering in geophysical systems represents an energetically-preferred state of the system that can be considered a manifestation of the minimum energy dissipation principle. Our analysis implies that, for a given pressure head, a more homogeneous porous matrix will result in less pronounced preferential flowpaths, along with lower flow and higher mineralization rates. On the other hand, for a highly heterogeneous matrix dominant preferential flowpaths will be obtained, along with higher flow and lower mineralization rates. Considering these aspects for carbon sequestration where acidified brine leads to carbon mineralization, we conclude that, for a given pressure head, an injection into a more heterogeneous matrix will result in a higher injection rate, while a more homogeneous domain will yield a higher mineralization rate, thus exemplifying the resulting trade-off in the injection strategy.*

**Appendix A: Non-equilibrium thermodynamics in a nutshell**

A *non-equilibrium system* is, as the name implies, a system that is not in thermodynamic equilibrium. In such a system gradients of the thermodynamic state properties such as temperature, pressure and concentration of chemical species

are present, as opposed to a system in equilibrium where these properties are homogeneous throughout. The presence of gradients in *thermodynamic potentials* such as pressure, temperature and chemical potential implies a net transfer of energy or matter, or *thermodynamic flux*, within the system or across its boundaries. Examples include heat transfer, where heat flux travels in the medium due to applied temperature gradient, hydrodynamic flow where mass flux is driven by hydraulic head gradient, mixing of chemical species and chemical reaction, both driven by the gradient of chemical potential (see left column in Figure 4). All of these processes may be seen as *dissipative*, meaning that the *free*, or useful

[Figure]

**Figure 4.** Examples of non-equilibrium processes (left column) and systems in equilibrium (right column) in (a) heat transfer and hydrodynamic flow, (b) mixing of chemical species and (c) chemical reaction.

energy in the system (energy available to perform useful work, for example internal, kinetic or potential) is being expended, since energy must be constantly supplied to the system in order to maintain these fluxes. Obviously, free-energy of the system is not a conserved property. A system that receives influx of energy from surroundings is said to be an open system, as it interacts with surroundings. Such a system may be maintained in a non-equilibrium thermodynamic stationary state of constant thermodynamic flux. On the other hand, a system without constant supply of energy from outside will soon deplete its available free-energy, reducing its capacity to do thermodynamic work and leading to a decline in the thermodynamic potential gradient and, therefore, cessation of the resulting thermodynamic flux. Such a system is found in a state of equilibrium, characterized by homogeneity in thermodynamic state properties (see right column in Figure 4). Thus, to maintain mass flux of a fluid through a pipe, a constant hydraulic head gradient must be

maintained between the pipe ends to overcome viscous frictional effects and gravitational head differences. This can be done by operating a pump that maintains the head gradient and, thus, the mass flow in a pipe by consuming electrical power. Should the supply of electrical power to the pump cease, the head gradient will no longer be maintained and, at some point, the mass flow through the pipe will stop, as the available free-energy in the system has been depleted by viscous dissipation. Thus, such a process is clearly a dissipative one.

An important outcome of non-equilibrium thermodynamics is that dissipative irreversible processes produce entropy at a rate that is directly related to the power dissipated during the process, the proportionality constant between them being the local temperature (see Section 3.1 in the manuscript). Here, an assumption of *local equilibrium* is required, implying that a non-equilibrium system still experiences equilibrium, albeit on a local scale. Thus, instead of a homogeneous temperature that characterizes system in equilibrium, in a non-equilibrium system we may assume that each small volume of the system is locally in equilibrium, thus temperature (and other thermodynamic state variables) can be defined locally. This results implies that, by studying entropy generation due to various dissipative processes pertinent to the non-equilibrium system under consideration, important observations can be made regarding the dynamics of its physical behavior. Thus, a decrease in entropy production means that the process now occurs in a more efficient way, with less useful energy depleted per unit of transferred thermodynamic flux.

When applying the non-equilibrium thermodynamic framework to the problem of reactive flow in porous media characteristic of geochemical weathering, where dissolution-precipitation of the porous matrix takes place, it is natural to concentrate on the following interrelated processes: (a) *percolation*, or fluid transport through the porous matrix, which involves dissipation of hydraulic power while overcoming the hydraulic resistance of the matrix due to viscous friction effects, affects concentration distribution of the chemical species and, therefore, directly influences chemical reaction, (b) *mixing* of chemical constituents that involves dissipation of chemical energy and affects the reaction rate and (c) *chemical reaction* of dissolution-precipitation, also involving dissipation of chemical energy, which affects directly the transport through matrix.

For further details regarding non-equilibrium thermodynamics see, for instance, Kondepudi and Prigogine (1998).

**Appendix B: Employing Shannon entropy to quantify transport self-organization**

To revise the basic concept of information entropy, let us consider a single probabilistic event, having a number of possible outcomes that are assumed equally probable. For example, in the case of a single symbol transmitted in the Morse code, we have two possible outcomes (realizations) for this event - a dash and a dot (omitting the intermission between the transmitted letters). Employing the indices 0 and 1 to denote parameters before and after the event realization, initially, we have $N_0 = 2$ equally possible outcomes of the message transmission event and zero initial information $I_0 = 0$.

Following the transmission of a single symbol, we have a single realization of the event - a dash or a dot, thus $N_1 = 1$, and a non zero information due to the receipt of a symbol $I_1 \neq 0$.

In case of a sequence of events enumerated $1...n$, having a respective number of outcomes $N_{0_1}, N_{0_2}, ..., N_{0_n}$ (consider a word that consists of $n$ symbols in the Morse alphabet), the total number of possible outcomes is $N = N_{0_1} \cdot N_{0_2} \cdot ... \cdot N_{0_n}$. Intuitively, we would expect the information measure to be an additive parameter so that the amount of information contained in a word transmitted with the help of the Morse code would be equal to the sum of the amounts of information for each symbol that constitutes the word, that is $I(N) = I(N_{0_1} \cdot N_{0_2} \cdot ... \cdot N_{0_n}) = I(N_{0_1}) + I(N_{0_2}) + ... + I(N_{0_n})$. This can be fulfilled by choosing $I = b \ln N$, where $b$ is an arbitrary constant parameter that amounts to a choice of a unit of measure (in fact, Shannon (1948) gave a formal proof that the logarithm function is the only possible relation between $I$ and $N$ that possesses, in addition to additivity, also continuity and monotonicity properties). For a binary system that consists of only two symbols, such as the dash and the dot in the Morse alphabet, a transmitted word of a total $n$ symbols has $N = N_{0_1} \cdot N_{0_2} \cdot ... \cdot N_{0_n} = 2^n$ possible realizations. Taking a single transmitted symbol as one unit of information, we demand that $I = b \ln N = n$ to obtain $b = \log_2 e$ and $I = \log_2 N$. Since a single position in a sequence of symbols in a binary system is defined as a bit, the corresponding units of information $I$ will be called $bits$ (an abbreviation from *binary digit*).

Now, assume that each of the final amount of different symbols that constitute a message has its own relative occurrence frequency (the respective probability of finding a specific symbol at a specific place in a sequence), such as the case of different letters in an alphabet. In the case of a Morse code, assume that the transmitted $n$-symbol word consists of $n_1$ dashes and $n_2$ dots, so that $n_1 + n_2 = n$. Using some results from combinatorics, it can be shown that

$$S = I/n = -\Sigma_i \, p_i \log_2 p_i \tag{1}$$

where $S$ is the information entropy (also referred to as the Shannon entropy) per symbol and $p_i = n_i/n$, $i = 1, 2$ are the relative occurrence frequencies of both symbols. Maximum Shannon entropy obtained during the transmission of a message is when the relative occurrence frequency of each symbol is identical $p = 1/2$. It is easily shown that this maximum value equals to $S_{max} = -\log_2 p = 1$. The relation (1) can be easily generalized for any number of possible outcomes per event. Thus, a definition of the information entropy is obtained that possesses an additivity property similar to physical properties such as entropy, energy, and mass. An example that may be seen as a consequence of the results obtained with the help of the information theory is that in modern communication methods, the more common alphabet letters are encoded in such a way that their information content is minimized in order to facilitate the transmission process. Thus, in a Morse code, a letter $E$ is encoded by a single dot, while $J$ is a dot followed by three dashes. For more details on the topic of Shannon entropy, see Shannon (1948).

This definition resembles the definition physical entropy in statistical mechanics, as defined by Gibbs, where the logarithm in Eq. (1) is to the base of $e$ and the sum is multiplied by the Boltzmann constant. The statistical definition of physical entropy characterizes the number of possible microstates of a system that are consistent with its macroscopic thermodynamic properties, which constitute the macrostate of the system. Thus, in the case of a gas consisting of a large number of molecules in a container, a microstate of the system consists of the position and momentum of each molecule as it moves within the container, colliding with other molecules and container walls. A multitude of such microstates correspond to a single macroscopic state of the system, defined by its pressure and temperature. The parameter $p_i$ in this case corresponds to a probability that a microstate $i$ occurs during the system's fluctuations. According to the second law of Thermodynamics, the entropy of an isolated system reaches its maximum value at equilibrium, where gradients of thermodynamic parameters are depleted by dissipative forces and the measure of order in the system is at its minimum. In this case, each microstate is equally likely and $p_i$ is simply the inverse of the total number of microstates (Kondepudi and Prigogine, 1998). Returning to the context of communication theory, maximum entropy is obtained during the transmission of a message of a length $n$ when an alphabet with the largest number of symbols is employed, and the relative occurrence frequency of each symbol is identical.

To characterize the emergence and evolution of transport self-organization in a computational field as the dissolution-precipitation reactive processes in the field advance, we adopt a straightforward use of the Shannon entropy, in a similar vein to Zehe et al (2021) and Shavelzon and Edery (2024), where it was employed in the context of characterizing self-organization in non-reactive flow in a heterogeneous porous medium. We calculate the Shannon entropy of particle distribution in the direction transverse to flow at a dimensionless time $\tilde{t}$ as a function of distance from the inlet $x$, $S(x, \tilde{t})$, using the same formula as in (1), reprinted here as

$$S(x, \tilde{t}) = -\Sigma_i p_i \log_2 p_i \qquad (2)$$

where $p_i = N_i / N_{tot}$, $i = 1 ... N_y$ is the relative concentration distribution at $x, \tilde{t}$ in the direction transverse to flow (here, $N_i$ is the current number of particles of a specific specie in the $i$-th cell in the vertical direction at $x$ and $N_{tot}$ is the total number of particles of that specie in all cells at $x$, taken at a dimensionless time $\tilde{t}$). This quantity has an upper bound of $S_{max} = \log_2 N_y$, corresponding to the state of maximum entropy, or homogeneity of transport in the field, where the same number of particles has visited each of the $N_y$ computational cells at a given distance from inlet $x$; the obvious lower bound of 0 corresponds to the state of maximum transport spatial organization in the field, namely all particles follow a single preferential flow path (Zehe et al (2021) and Shavelzon and Edery (2024)).

**Appendix C: 2. Methodology - modeling the reaction-transport interaction, representative of chemical weathering**

The dynamics of the reaction-transport interaction in the heterogeneous porous calcite matrix is investigated using a 2D computational field that represents a sample of the calcite mineral of dimensions $12 \times 30 [\text{cm}^2]$, initially in chemical

equilibrium with the resident water of pH 8. The field is represented by the hydraulic conductivity distribution $K$ and is discretized into $150 \times 60$ computational cells, each having the size of $\Delta x, \Delta y = 0.2[\text{cm}]$. Low-pH water (pH of 3.5) is injected at the inlet, disrupting the existing chemical equilibrium and reacting with the porous calcite matrix. Hydraulic conductivity fields of various degrees of heterogeneity, defined by the prescribed values of conductivity variance $\sigma_0^2$ and spatial correlation length $l_c$, are generated using the SGSIM sequential Gaussian simulator (Deutsch (1997)). Varying $\sigma_0^2$ values are considered, while $l_c$ is kept constant such that the ratio $\Delta x/l_c = 4$ is obtained (here the subscript $0$ denotes parameters referring to the initial hydraulic conductivity distribution, prior to the occurrence of reaction). This is considered a good enough discretization to capture the bigger-scale structures in the heterogeneous field and their effect on the statistics of the coupled transport-reactive process. The field realizations obtained from SGSIM are considered as distributions of the natural logarithm of conductivity, thus the hydraulic conductivity field is obtained by taking their natural exponent. Constant initial porosity value of $\theta_0 = 0.43$ is assumed for each field. We apply a hydraulic pressure head drop boundary condition $\Delta h_{BC}$ between the inlet and outlet of the field, while the horizontal boundaries are represented by reflective walls. For a detailed description of the computational setup and its validation, refer to Shavelzon and Edery (2024).

**2.1 Flow and transport in porous matrix**

Upon applying the hydraulic head drop boundary condition to the porous calcite matrix, flow commences in the matrix. The distributions of the hydraulic pressure head $h$ and the water Darcy flux $\mathbf{q}$ in the computational field as functions of the spatial coordinate $\mathbf{x}$ are governed by the continuity equation and the Darcy's law (3)–(4), which are solved using Finite element method (Guadagnini and Neumann (1999)). They are then used to calculate the flow velocity field $\boldsymbol{v}$, required to simulate the reactive transport in the field.

$$\nabla \cdot \mathbf{q}(\mathbf{x}) = 0 \tag{3}$$

$$\mathbf{q}(\mathbf{x}) = -K\nabla h(\mathbf{x}) \tag{4}$$

Throughout the paper, vector fields will be specified using bold font, while scalar fields will appear in the regular font. We employ the Lagrangian particle tracking approach to simulate the solute transport in the field (Le Borgne (2008)). The invading low-pH water is represented by the reactive particles injected at a constant rate into the porous matrix, flux-weighted according to the conductivity distribution of the inlet cells. The Langevin equation is employed to advance the injected particles in the field. The general Langevin equation for a stochastic vectorial variable $\mathbf{X}$ has the form (Risken (1996))

$$\frac{d\mathbf{X}}{dt} = \boldsymbol{h}(\mathbf{X}, t) + g(\mathbf{X}, t)\boldsymbol{\Gamma}(t) \tag{5}$$

where $\boldsymbol{h}(\mathbf{X}, t), g(\mathbf{X}, t)$ are functions of $\mathbf{X}$ and the computational time $t$ (not to confuse with the hydraulic head distribution $h$), and $\boldsymbol{\Gamma}(t)$ is a vectorial Gaussian random variable with $\Gamma^i(t)$ independent and standard-normally distributed.

To simulate the trajectory of a solute particle moving in the porous calcite matrix $\boldsymbol{X}(t)$, affected by both the advective velocity field $\boldsymbol{v}$ and the diffusive transport process with diffusion coefficient $D$, we set $\boldsymbol{h} = \boldsymbol{v}$, $g = \sqrt{2D}$ (Perez (2019)). Here, $\mathbf{X}(t)$ is the position of a particle, $\boldsymbol{v}$ is the flow velocity field, obtained from (3)–(4), $t$ is the computational time and $D$ is the diffusion coefficient of the injected fluid. For the diffusion coefficient, a representative value of $D = $1e-5 $[\mathrm{cm}^2\,\mathrm{min}^{-1}]$ was employed (Domenico and Schwartz (1997)). To be solved numerically, we discretize (5) using the Euler-Maruyama method (Kloeden (1992)). We employ the *ran2* random Gaussian generator (Press(1992)) for the stochastic variable generation.

The Lagrangian particle tracking simulation has been validated using the scenario of a 2D instantaneous injection in a homogeneous porous medium, governed by a steady state flow. For validation, we used the equivalence property of a statistical ensemble governed by the Langevin equation (5) to the advection-diffusion equation (Risken (1996), Perez (2019)), whose solution for this scenario constitutes a traveling Gaussian wave (Kreft (1978)). See Appendix in Shavelzon and Edery (2024).

**2.2 Chemical reaction model**

We simulate the complex dynamics of chemical weathering by modeling the reaction-transport interaction in a heterogeneous porous calcite matrix, initially in chemical equilibrium with the resident water of pH 8. Low-pH water (pH of 3.5) is injected at the inlet, disrupting the existing chemical equilibrium and reacting with the porous calcite matrix. Upon entering the calcite matrix, the invading low-pH water causes dissolution of calcite, along with the production of dissolved calcium and carbonic acid, as shown by the following reactions (Singurindy and Berkowitz (2003), Edery et al. (2011))

$$CO_3^{2-} + 2H^+ \leftrightarrow H_2CO_3 \tag{6}$$
$$CaCO_{3(s)} \leftrightarrow CO_3^{2-} + Ca^{2+} \tag{7}$$

Here, two hydrogen ions H+, associated with the injected low-pH water, combine with CO3(2-) to produce H2CO3, thus causing dissolution of the calcite porous matrix in accordance with the Le Chatelier principle. The deprotonation reaction (3) represents the sum of two equilibria of carbonic acid (Manahan, 2000). The pH level of the fluid is assumed to be bounded by that of the invading (pH 3.5) and the resident fluid (pH 8). This simplified chemical representation constitutes the specific case of Edery et al. (2011), where de-dolomitization was also included in the chemical model, and was treated extensively in Edery et al. (2021) and Shavelzon and Edery (2024) (see also Singurindy and Berkowitz (2003), Al-Khulaifi et al (2017)). While simplified, this chemical reaction setup nevertheless captures the qualitative dynamics of the complex process of geochemical weathering, making it possible to analyze the complex reversible behavior using a relatively unsophisticated model.

We assume an abundance of dissolved Ca2+, thus making the reaction (4) non-rate-limiting (this assumption is not that far-fetched in karst systems - see, for instance, Amorsson (1981)). Therefore, reaction (3) becomes rate-limiting, and the reactive process is controlled by the available H+, in consistency with observations from previous studies (Edery et al. (2011), Singurindy and Berkowitz (2003)). The spatial distribution of H+ in the calcite matrix is, in turn, controlled by transport and reaction processes. Combining reactions (3)-(4) along the lines of Edery et al (2021), we obtain the following simplified equation that represents an interaction between hydrogen ions, carbonic acid and calcite

$$a = h + c \tag{8}$$

where $a, h, c$ denote $H_2CO_3, 2H^+, CaCO_3$, respectively. The dissolution-precipitation reaction is fully reversible, with the opposite precipitation reaction possible under favorable pH conditions. The direction of reaction is governed by the deviation from the chemical equilibrium, defined by the current concentrations of H+ and H2CO3 in the localized area of reaction: dissolution will occur in the presence of H2CO3 concentration above the equilibrium value for the current pH and vice versa, based on the Bjerrum plot for carbonic acid dissociation (Manahan (2000)). For a detailed description of the computational setup and its validation, refer to Edery and Berkowitz (2011), Edery et al (2021) and Shavelzon and Edery (2024).

**2.3 Chemical kinetics and reaction-transport interaction**

In our reactive transport model the kinetics of the reactive process operate accordingly to an algorithm, developed to mimic the actual chemical dissolution-precipitation reactions. At $t = 0$, $H^+$ particles, that represent the low-pH water, are being injected into the field at a constant rate and advanced in the field in accordance with (5) for the duration of the computational time step $\Delta t$. Each $H^+$ particle represents a molar parcel of hydrogen ions (see Shavelzon and Edery (2024) for the exact procedure). Then, reaction occurs as described by (8), during which some of the $H^+$ particles transform into $H_2CO_3$ particles. According to (8), each newly created $H_2CO_3$ particle receives half of this amount of carbonic acid. We assume that the characteristic reaction timescale is negligible with respect to the transport timescale, thus the reaction is instantaneous and proceeds until chemical equilibrium is reached everywhere in the field. This corresponds to an infinite local Damkohler number.

For equilibration purposes, the amount of $H^+$ and $H_2CO_3$ in each computational cell is assessed based on the current number of particles of both types in the cell. The current pH level in a cell is estimated based on the number of $H^+$ particles there and is employed to calculate the fractional amount of $H_2CO_3$ in equilibrium $\alpha_1(pH)$ as

$$\alpha_1(pH) = \frac{10^{-2pH}}{10^{-2pH} + 10^{-pk_{a1}} + 10^{-pk_{a2}} + 10^{-pH-pk_{a1}}} \tag{9}$$

with $pk_{a1} = 6.35, pk_{a2} = 10.33$ (see Manahan (2000) for related data, as well as for the Bjerrum plot for carbonic acid dissociation). We then compare the value of $\alpha_1(pH)$ to the current fractional amount of $H_2CO_3$ in a cell. The reaction

proceeds in the direction established corresponding to the Le Chatelier principle for (8): we expect dissolution in case when the fractional amount of $H_2CO_3$ is smaller than its value in equilibrium $\alpha_1(pH)$, and vice versa. Equilibration in a cell consists of a series of reactive events, during each of which a single $H^+$ particle transforms into a $H_2CO_3$ particle or vice versa. Current values of pH level and the fractional amount of $H_2CO_3$ in a cell are recalculated following each event and the process is repeated until equilibrium in a cell is reached. In the current simulation, an error of 5 % in establishing the chemical equilibrium in a cell was considered acceptable.

Upon establishing equilibrium, we update the cell porosity $\theta$ as shown in (10), where the subscripts $k+1$ and $k$ denote porosity values prior to and after the reaction, $d[H_2CO_3]$ is the total increment of carbonic acid in the cell due to equilibration in the current time step (accepts negative values as well), $M_{CaCO_3} = 36.93$ [$\mathrm{cm}^3\,\mathrm{mol}^{-1}$] is the molar volume of calcite (Morse and Mackenzie (1990)) and $\Delta x \Delta y$ is the cell volume assuming unit depth. We then employ the Kozeny-Carman relation (11) to update the hydraulic conductivity $K$ value in a cell

$$\theta_{k+1} = \theta_k + \frac{d[H_2CO_3]M_{CaCO_3}}{\Delta x \Delta y} \tag{10}$$

$$K_{k+1} = K_k \cdot \frac{\theta_{k+1}^3}{(1-\theta_{k+1})^2} \cdot \frac{(1-\theta_k)^2}{\theta_k^3} \tag{11}$$

This process is repeated for all computational cells before injecting new particles and advancing all existing particles again during the next time step $\Delta t$. The hydraulic pressure head distribution $h$ is recalculated for the reaction-altered hydraulic conductivity field from (3)–(4) at constant time intervals of $10\Delta t$, thus introducing the reaction-transport interaction in the computational model. Reaction enhancement is employed in the simulation by magnifying the cell porosity increment due to reaction (10) by a certain factor, thus accelerating the reaction-transport interaction in the field, while leaving unamended the overall reaction-transport dynamics. This allows reduction of computational costs (Shavelzon and Edery (2024)). The numerical simulation proceeds until the time equal to twice the pore volume time $T_{pv}$ is reached. $T_{pv}$ is determined for each SGSIM-generated realization of hydraulic conductivity $K$ using the non-reactive particle tracking algorithm (NRPT), where the particles are being injected at the inlet and traverse the field while omitting the reaction part, and is set to be the time when $50\%$ of the injected particles have reached the outlet.

**References:**

– Shavelzon and Edery (2024), Shannon entropy of transport self-organization due to dissolution–precipitation reaction at varying Peclet numbers in initially homogeneous porous media,

– Shannon (1948), A Mathematical Theory Of Communication.

– Zehe et al (2021), Preferential Pathways for Fluid and Solutes in Heterogeneous Groundwater Systems.

– Edery and Stolar (2021), Feedback mechanisms between precipitation and dissolution reactions across randomly heterogeneous conductivity fields.

- Edery et al (2011), Dissolution and precipitation dynamics during dedolomitization.

- Singurindy and Berkowitz (2003), Flow, dissolution, and precipitation in dolomite.

- Al-Khulaifi et al(2017), Reaction Rates in Chemically Heterogeneous Rock: Coupled Impact of Structure and Flow Properties Studied by X-ray Microtomography

550
- Manahan (2000), Environmental Chemistry (9th edn).

- Amorsson (1981), Mineral Deposition From Icelandic Geothermal Waters: Environmental and Utilization Problems.

- Snaebjornsdottir et al (2020), Carbon dioxide storage through mineral carbonation.

- Ruprecht and Falta (2012), COMPARISON OF SUPERCRITICAL AND DISSOLVED CO2 INJECTION SCHEMES.

555
- Pradhan et al (2025), Determination of CO2 solubility in brines and produced waters of various salinities for CO2 EOR and storage applications.

- Ahmad et al (2016), Injection of CO2-saturated brine in geological reservoir: A way to enhanced storage safety.

- Oldenburg et al (2001), Carbon Sequestration with Enhanced Gas Recovery: Identifying Candidate Sites for Pilot Study.

560
- Cao et al (2013), Dynamic alterations in wellbore cement integrity due to geochemical reactions in CO2-rich environments.

- Haugen et al (2023), Multi-scale dissolution dynamics for carbon sequestration in carbonate rock samples.

- Harrison et al (2015), The impact of evolving mineral–water–gas interfacial areas on mineral–fluid reaction rates in unsaturated porous media

565
- Kondepudi and Prigogine (1998), Modern Thermodynamics: From Heat Engines to Dissipative Structures.

- Kaufmann et al (2016), Dissolution and precipitation of fractures in soluble rock

- Szawello et al (2024), Quantifying dissolution dynamics in porous media using a spatial flow focusing profile.

- Liu et al (2005), Dissolution-induced preferential flow in a limestone fracture.

- Jones and Detwiller (2016), Fracture sealing by mineral precipitation: The role of small-scale mineral heterogeneity.

570
- Kohlhaas et al (2025), Numerical Investigation of Preferential Flow Paths in Enzymatically Induced Calcite Precipitation supported by Bayesian Model Analysis.

- Deutsch (1997), GSLIB Geostatistical Software Library and User's Guide.

- Morse and Mackenzie (1990), Geochemistry of Sedimentary Carbonates

575
- Kreft and Zuber (1978), On the physical meaning of the dispersion equation and its solutions for different initial and boundary conditions.

– Perez et al (2019), Reactive Random Walk Particle Tracking and Its Equivalence With the Advection-Diffusion-Reaction Equation.

– Risken (1996), The Fokker-Planck Equation: Methods of Solution and Applications.

580

– Press et al (1992), Numerical Recipes in C: The Art of Scientific Computing.

– Kloeden (1992), Numerical solution of stochastic differential equations.

– Domenico and Schwartz, Physical and Chemical Hydrogeology, 2nd Edition.

– Le Borgne et al (2008), Lagrangian Statistical Model for Transport in Highly Heterogeneous Velocity Fields.

– Gudagninin and Neumann (1999), Nonlocal and localized analyses of conditional mean steady state flow in bounded,

585

randomly nonuniform domains: 1. Theory and computational approach.

---

## Author Response (AR1)

**Linking chemical weathering, evolution of preferential flow paths and transport self-organization in porous media using non-equilibrium thermodynamics**

Evgeny Shavelzon1, Erwin Zehe2, and Yaniv Edery1

**Correspondence:** Yaniv Edery(yanivedery@technion.ac.il)

**Authors' replies to Reviewers**

This document contains replies to Reviewers' comments. We appreciate the time and effort invested by the respectable Reviewers in our manuscript and are grateful for the opportunity to resubmit a revised draft for your consideration. We address all Reviewers' comments in the following itemized list, arranged such that the answer to each review item is located directly below the item. Please note that Reviewers' comments and our responses are given in black and blue, respectively.

All proposed changes have been implemented in the updated manuscript. Due to suggested major revision, significant parts of the manuscript were rewritten or significantly updated. Therefore, in the attached Track changes document the changes are marked in blue. Sincerely, Evgeny Shavelzon, Erwin Zehe and Yaniv Edery.

**Replies to Reviewer 1**

**Reviewer 1**

This manuscript is about the dissolution of calcite in porous media and the resulting self-organization of preferential flow paths. The results are based on numerical simulations and are analyzed with the help of thermodynamic variables. Although the basis of the model seems to be similar to what has been assumed in models of karst evolution for decades, I found this study very interesting. The are numerous points to think about. On the other hand, I find it a bit too difficult. Each step is not really difficult in itself, but in sum it is very challenging not to get lost. When reading it, I had to go back quite often to find out whether something is indeed not clear or whether I just missed it. So, I am a bit afraid that scientists who did not have to hold on to write a review will likely give up.

<sup>1Faculty of Civil and Environmental Engineering, Technion - Israeli Institute of Technology, Haifa, Israel

<sup>2Karlsruhe Institute of Technology (KIT), Institute of Water and River Basin Management, Karlsruhe, Germany

We thank the Reviewer for their careful review and positive comments that were extremely helpful in clarifying and improving the readability of the manuscript. Indeed, as the reviewer has pointed out, the main point of the study is the analysis of the process of geochemical weathering within thermodynamic framework and establishing the correlation between chemical weathering and the emergence of preferential flowpaths in the system, rather than the modeling of the geochemical process itself, which has been done previously. We are thankful to the reviewer for indicating the need to improve the readability of the manuscript, clarifying it and making it more concise. We fully acknowledge this need and are committed to amending the manuscript accordingly. Please find attached the summary of our suggestions regarding improving the readability towards the end of our reply.

So let me just make some suggestions, hoping that they may be helpful for clarifying some aspects. Quite in the beginning, I wondered about the porosity and permeability (Eqs. 7 and 8). If I got it correctly, you assume constant initial permeability and introduce spatial variability be assuming a variable factor in the Kozeny-Carman relation. This looks unconventional. The conventional approach would be the same factor, but variable initial porosity. I guess that you used this approach in order to define the pore volume time. However, I think that it also has an effect on how the development of the permeability. In your approach, a given change in porosity (from the constant initial value) introduces the same relative change in permeability. In the more conventional approach, the relative change would be smaller for cells that already have a high permeability. So I would guess that the formation of preferential flow paths in stronger in your approach than in the conventional version. This makes we wonder whether it has an effect on the results. Maybe you can clarify this aspect.

We thank the reviewer for addressing this important point. The constant initial porosity assumption was introduced in order to facilitate the chemical aspects of our reactive transport model, namely the calculation of the molar amount of  $H^+$  to give pH of 3.5 in a computational cell. This allows to estimate the current pH in the cell and, thus, the direction and the extent of reaction there (see Section 2.3 in Shavelzon and Edery (2024) for details). This assumption was deemed acceptable, as the purpose of our simplified chemical reaction setup is to capture the qualitative dynamics of the complex process of geochemical weathering, thus making it possible to simulate and analyze the complex reversible behavior using a relatively unsophisticated model.

In our model, the change in porosity of a specific cell in the computational field is determined by the amount of reaction that has occurred in a cell during the current time step, as dissolution/precipitation of calcite increases/reduces the void volume of the cell (as expressed by Eq.(9) in the *updated* manuscript). Given the initial and the updated values of porosity in the cell, as well as the initial permeability value, we are then able to calculate the updated permeability in the cell using the Kozeny-Carman relation as shown in Eq. (10) in the updated manuscript. Notice, that Eq. (10) includes both initial and updated porosity values in a nonlinear relation that cannot be expressed as a function of only a change in porosity. Therefore, the relative change in permeability does depend on the initial porosity value. To demonstrate this, we attach below the plot of the Relative change in permeability as a function of porosity  $K_{k+1}/K_k$ , as given by the expression in Eq. (10) as a function of initial porosity  $\theta_k$ , for the change in porosity  $\Delta\theta = 0.05$  - see Figure 1 in the current document. We observe that for larger  $\theta_k$  values, that correspond to higher permeability, the relative change in permeability is smaller, just as is physically expected. In our model, we make a simplifying assumption of constant

**Figure 1.** Relative change in permeability as a function of initial porosity.

initial porosity value of 0.43 for all field realizations, regardless of their heterogeneity degree, while the permeability is distributed randomly based on prescribed values of distribution variance and correlation length as explained in Section 2.1 (lines 150–156) in the updated manuscript. While this assumption cannot be seen as rigorous, we argue that it doesn't impact significantly the simulation results, as well as the main conclusions made, since the main point of the manuscript is to capture the essential behavior of the complex coupled process of geochemical weathering, simple enough to not make the reactive model the main issue of the manuscript, but sufficient to be able to draw conclusions about the implications of the ensuing reaction-transport interaction.

We propose to add the following paragraph to the manuscript to clarify this point (lines 156–160 in the updated manuscript):

The constant initial porosity assumption, introduced in order to facilitate the chemical aspects of our reactive transport model, was deemed acceptable, as the purpose of our simplified chemical reaction setup is to capture the qualitative dynamics of the complex process of geochemical weathering, thus making it possible to simulate and analyze the complex reversible behavior using a relatively unsophisticated model.

– My main concern is, however, that the entropies are somewhat abstract properties.

We are sincerely thankful to the Reviewer for pointing out the need to clarify the meaning of both the Shannon and the physical entropies that are employed extensively in the manuscript. In the manuscript, we are looking for the correlation between the emergence and intensification of preferential flowpaths and the dissipation of the useful energy of the system due to different processes, characteristic for chemical weathering, such as chemical reaction, mixing of chemical species and percolation (frictional dissipation due to fluid viscosity). To quantify the intensity of the preferential flowpaths phenomenon (how much is the transport concentrated, or self-organized, within these paths, as opposed to a homogeneous distribution throughout the field), we employ the Shannon entropy, or information entropy – a quantity borrowed from information theory and used, among others, to quantify self-organization in physical systems (see the original paper on this subject by Shannon (1940), as well as Zehe et al (2021) and Shavelzon and Edery (2024) for application to transport self-organization in porous media). On the other hand, to quantify the useful energy dissipation during chemical weathering, we employ the physical entropy, a quantity that, within the framework of nonequilibrium thermodynamics, allows to quantify energy dissipation due to chemical reaction, mixing of chemical species and percolation – see, for instance, Kondepudi and Prigogine (1998). Thus, by employing quantities from seemingly unrelated disciplines, we attempt to correlate between these two parameters in order to obtain a deeper understanding of the phenomenon of preferential flowpaths. These quantities, however, are far from being abstract, but have a precise physical meaning. The main conclusion we arrive at after analyzing both the Shannon entropy of transport self-organization and the physical entropy sources is that a clear correlation is established between the emergence and intensification of preferential flowpaths and the useful energy dissipation due to reaction, mixing and percolation in geochemical systems that involve chemical weathering. The emergence of preferential flowpaths allows the system to reduce this useful energy dissipation rate and, thus, represents an energetically preferred state of the system.

To improve the clarity of presentation of a non-equilibrium thermodynamic framework in the manuscript, we propose adding a short chapter that explains in simple terms the topic of dissipative processes in non-equilibrium thermodynamics to the Supplementary. See **Supplementary 2. Non-equilibrium thermodynamics in a nutshell** in the updated manuscript.

We also propose adding a short section that briefly explains the concept of the Shannon entropy and its use for quantifying self-organization in our physical scenario, along the lines of Section 3.2 and Supplementary 2 in Shavelzon and Edery (2024) (currently, the manuscript draft references this publication, where this subject is treated extensively). See

Section 3.2 in the *updated* manuscript (lines 371—401). Also, notice that references to Supplementary 2 in Shavelzon and Edery (2024), which contains a brief background on the Shannon entropy, are given in the appropriate places in the manuscript. In that way, both the Shannon and the physical entropy, as well as their relevance in our manuscript, are explained in the updated manuscript before any use of them is implemented in the Results. Full text of Supplementary 2 in Shavelzon and Edery (2024) is given in the **Appendix A: Employing Shannon entropy to quantify transport self-organization** in the current document for the reviewer's convenience.

- It all makes sense, but I am still left with the feeling that I do not understand fully what happens in the system. One point is why H+ is associated with the preferential flow paths more closely than H2CO3.

We are sincerely thankful to the Reviewer for pointing out this subject, that has not received a sufficient explanation in the manuscript. Different spatial patterns for dissolution and precipitation reactions, associated with an abundance of H+ and H2CO3 respectively, in reactive flow in heterogeneous / fractured porous media have been reported both experimentally (Liu et al (2005), Jones and Detwiller (2016)) and numerically (Edery et al (2021), Kohlhaas et al (2025), Kaufmann et al (2016), Szawello et al (2024)). For dissolution-dominated flows, such as in our case, dissolution occurs mostly within the preferential flowpaths, characterized by higher permeability and flow velocity and, thus, ample supply of reactant (these paths act as main channels for reactant supply). Higher flow velocity within the paths limits the residence time and, thus, the amount of precipitation within the path itself (which is also limited by the more uniform transport within the path that limits mixing, therefore allowing for less chance to disrupt the local chemical equilibrium). However, at the periphery of the paths, which is less permeable as a rule, residence time increases as well as the chance to disrupt local chemical equilibrium, thus leading to increased precipitation there. Both these processes further enhance the preferential flowpaths by making the paths themselves more permeable by virtue of excessive dissolution there, thus attracting even more reactant within the paths, along with making the periphery of the paths even less permeable via an increased precipitation there. This preferential flowpaths enhancement leads to transport self-organization, or channelization, as attested by the decrease in Shannon entropy over time (see Figure 5 in the updated manuscript).

In our manuscript, we simulate the dissolution-dominated reactive flow scenario by injecting low-pH water into the calcite porous matrix. In our chemical reaction model, dissolution of calcite is associated with an abundance of hydrogen ion H+, therefore it is reasonable that it will self-organize within the preferential flow paths, dominated by the dissolution reaction. On the other hand, H2CO3, responsible for precipitation in our model, is expected to be less associated with the preferential flowpaths but rather to exhibit a more homogeneous distribution due to increased precipitation in the vicinity of these paths. Following the respectable Reviewer's comment, we propose to amend the manuscript by adding this discussion in Results/Conclusions (see lines 485–498 and 688-692 in the updated manuscript).

 In this context, I would find it very helpful to separate the propagation of the reaction front from the changes in the flow pattern. So considering the dynamics of the reaction without taking into account the effect on the permeability. Then it would be easier to understand what exactly the effect of the changes in permeability is. Maybe the dynamics of the reaction is much faster and it is already clear, but I am not sure.

We sincerely thank the reviewer for suggesting ways to clarify the manuscript. However, we would like to draw the reviewer's attention to the fact that the main point of the manuscript is is the specific interaction between the reactive and transport processes in the calcite porous matrix that leads to the emergence of preferential flowpaths in the matrix. Thus, we will not gain any additional insight by examining the reaction separately from the changes in permeability, as within the particle tracker simulation we cannot account for the reaction without altering the permeability in response to it, since the dissolution-precipitation reaction alters the transport properties of the porous matrix. On the contrary, we aim at showing how the reaction-transport interaction affects both of these processes: the dissolution of calcite, that occurs in the preferential flow paths, intensifies these paths by making them more conductive (as shown by the decrease in Percolative entropy which signifies the decrease in frictional dissipation as the fluid makes its way through the porous matrix) and, therefore, increases the channelization of transport in these paths (as shown by the decreasing Shannon entropy of transport self-organization). On the other hand, this channelization of transport inhibits mixing of chemical species, therefore reducing the overall reaction rate (as shown by Reactive entropy). We are committed to make necessary amendments to Results/Discussion in manuscript to clarify this aspect (see lines 626-637 and 693-704 in the updated manuscript).

- Perhaps it would also be helpful to complement the 2D pictures of the concentrations in Fig. 3 by longitudinal profiles (integrated along the y-axis) in order to see whether the species are just concentrated more or less along the preferential flow paths or also follow strongly different distributions along the x-axis.

We thank again the reviewer for suggesting ways to further clarify the manuscript. The concentration of species along the preferential flowpaths, or, as we refer it in the paper, the intensity of the preferential flowpaths phenomenon (meaning how much is the spatial distribution of the species concentrated, or self-organized, within these paths, as opposed to a homogeneous distribution throughout the field), is quantified in the manuscript using the Shannon entropy (see explanation on this quantity in the text above, as well as in the reply to the last item in the document). These trends are shown in Figure 5 in the updated manuscript: the frames (a)–(b) show the Shannon entropy of spatial distributions of H+ and H2CO3 as a function of distance from inlet at a specific time, while the frames (c)–(d) show the mean value of Shannon entropy of spatial distributions of H+, H2CO3 and the total population over the field as a function of time. In general, the decrease in Shannon entropy of some distribution means an increase in its spatial organization (Zehe et al (2021), Shavelzon and Edery (2024)). The decrease in Shannon entropy with distance from inlet, as well as with an increase in variance of spatial permeability distribution  $\sigma_0^2$ , shows an increase in self-organization of species, meaning that their spatial distribution becomes less homogeneous, and, thus, more of the species find themselves within the preferential flowpaths. The decrease in Shannon entropy with time also shows an increase in self-organization of species as the time passes, as a result of reaction-transport interaction. We are committed to make necessary amendments to the manuscript to clarify this aspect (see lines 392-401 in Section 3.2 of the updated manuscript, as well as Supplementary 2 in Shavelzon and Edery (2024) - also reprinted in Appendix A in the current document).

- In summary: really interesting stuff, nothing wrong as far as I can see, but the paper would benefit from a more basic and clearer explanation of the model's behavior.

We thank the reviewer for expressing interest in the results presented in the manuscript, as well as for indicating the need to improve the readability of the manuscript, clarifying it and making it more concise. We fully acknowledge this need and are committed to amending the manuscript accordingly. Please find attached the summary of our suggestions regarding improving the overall clarity and readability of the manuscript, organized by chapter. All proposed changes have been implemented in the updated manuscript.

**(a) Abstract:**

Rewritten to better convey the key message of the manuscript regarding the correlation between the transport self-organization and the accompanying dissipative dynamics during chemical weathering, as well as to emphasize the geophysical aspects of chemical weathering and the implications of our study for real-world geophysical scenario of subsurface CO2 sequestration by mineralization (see lines 1–21 in the updated manuscript):

**Abstract.** Chemical weathering of soil and rock is a complex geophysical process during which the reaction and transport processes in the porous medium interact, causing erosion of the medium. This process is ubiquitous in geophysical systems and can be encountered, among others, in formation of karst systems, subsurface carbon sequestration and surface weathering of river beds. A common outcome of chemical weathering is the emergence and intensification of preferential flow paths, where the weathering alters the transport properties of the rock, thus creating coupling between transport and reaction. While numerous approaches have been undertaken to simulate this complex interaction, still a need exists for a unified framework able to correlate the emergence of preferential flow paths due to reaction-transport interaction with the associated dissipative dynamics. Here we propose such a framework considering the case of subsurface chemical weathering of calcite porous rock undergoing reversible dissolution-precipitation reaction, and apply non-equilibrium thermodynamics to analyze the ensuing reactiontransport interaction in this geophysical scenario. We identify the entropy generation sources, attributed to the dissipative processes inherent to this physical scenario and show a clear correlation between the emergence and intensification of preferential flow paths and the accompanying dissipative dynamics, where the evolution of the emerging paths leads to a decrease in the free-energy dissipation rate due to flow percolation, mixing of chemical constituents and reaction. This indicates that the emergence of preferential flow paths due to chemical weathering in geophysical systems represents an energetically-preferred state of the system that can be considered a manifestation of the minimum energy dissipation principle. Our analysis implies that, for a given pressure head, a more homogeneous porous matrix will result in less pronounced preferential flow paths, along with lower flow and higher mineralization rates. On the other hand, for a highly heterogeneous matrix dominant preferential flow paths will be obtained, along with higher flow and lower mineralization rates. Considering these aspects for carbon sequestration where acidified brine leads to carbon mineralization, we conclude that, for a given pressure head, an injection into a more heterogeneous matrix will result in a higher injection rate, while a more homogeneous domain will yield a higher mineralization rate, thus exemplifying the resulting trade-off in the injection strategy.

**215 (b) Chapter 1 (Introduction):**

Rewritten to improve clarity of the exposition: beginning from Viewing chemical weathering as a non-equilibrium thermodynamic process, we move to Overall review of analysis of non-equilibrium systems and then to Self-organization as a manifestation of non-equilibrium state. We proceed with Application of non-equilibrium thermodynamics to transport in porous media and conclude with Objectives of our study. See updated manuscript.

**(c) Chapter 2 (Methodology):**

- A conceptual flowchart added at the beginning of Chapter 2 (Methodology) see Figure 2 in the current document, as well as Figure 1 in the *updated* manuscript. This flowchart details the various stages of the employed methodology, from creating a realization of hydraulic conductivity field to numerical simulation to calculating the Shannon entropy and the physical entropy sources employing data obtained from simulation. The flowchart also presents graphically the research question of the manuscript, that concerns the correlation between the emergence of preferential flow paths (transport self-organization quantified using Shannon entropy) and the free-energy dissipation due to flow percolation, mixing and reaction (physical entropy sources). Each graphical block (shown by different colors in the flowchart) references the specific section where it is explained in details (section numbers have been adjusted to reflect the suggested manuscript amendments in the updated manuscript). See lines 137–142 in the updated manuscript.
- Section 2.2 (Chemical reaction model) rewritten and expanded to allow for a clearer description of the chemical model and its underlying assumptions. Also, we suggest minor corrections/additions to Chapter 2 as a whole to improve the overall presentation of methodology. See our suggestion for the updated Chapter 2 in the updated manuscript, specifically Section 2.2 (lines 193-221).
- A short section on the Lagrangian particle tracking model validation in 2D added (in Shavelzon and Edery, 2024 the validation was done for the 1D scenario) - see Supplementary 1 in the updated manuscript.
- Abstract terminology throughout the manuscript simplified where possible such as the Langevin equation paraphernalia (lines 77–85 in the updated manuscript) as related to purely technical aspects of our study and unnecessary for conveying the key message of the manuscript.

**(d) Chapter 3 (Non-equilibrium thermodynamic framework):**

Figure 2. Roadmap chart for the research procedure.

- Section 3.2 (Calculation of entropy generation terms) moved to Supplements, as it basically contains technical details that are not critical for understanding of the manuscript (See Supplementary 3 in the updated manuscript).
- A short chapter that explains in simple terms the topic of dissipative processes in non-equilibrium thermodynamics added to the Supplementary. See Supplementary 2 in the updated manuscript, also Section 3.1 there (lines 282–311).
- A short section added that briefly explains the concept of the Shannon entropy and its use for quantifying self-organization in our physical scenario along the lines of Section 3.2 and Supplementary 2 in Shavelzon and Edery (2024) (currently, the manuscript draft references this publication, where this subject is treated extensively). See Section 3.2 in the updated manuscript (lines 371—401). Also, notice that references to Supplementary 2 in Shavelzon and Edery (2024), which contains a brief background on the Shannon entropy, are given in the appropriate places in the manuscript. In that way, both the Shannon and the physical entropy, as well as their relevance in our manuscript, are explained before any use of them is implemented in the Results. Full text of Supplementary 2 in

Figure 3. Sources of inflow and dissipation of free energy in subsurface reactive flow.

Shavelzon and Edery (2024) is given in the **Appendix A: Employing Shannon entropy to quantify transport self-organization** in the current document.

- Figure that shows graphically the sources of inflow and dissipation of free energy in subsurface reactive flow replaced with an improved version that explains more clearly the sources of free-energy inlfow into the system and its dissipation (see Figure 3 in the current document, as well as Figure 2 in the *updated* manuscript). This figure visualizes the main result of Section 3.1: entropy generation, which is directly related to dissipation of the free-energy of the system, occurs in dissipative processes where the gradient of a thermodynamic potential drives the thermodynamic flux. Thus, the gradient of hydraulic head drives the mass flux, the gradient of the chemical potential drives the molar fluxes of the chemical species and the partial molar Gibbs energy drives the extent of reaction. See Section 3.1 (lines 283–310) and Supplementary 2 in the updated manuscript for detailed explanation.
- Abstract terminology throughout the manuscript simplified where possible such as differential operators used in the exposition of non-equilibrium thermodynamic framework (lines 345–360 in the updated manuscript) as related to purely technical aspects of our study and unnecessary for conveying the key message of the manuscript.

**(e) Section 4 (Results):**

- Shortened by (a) moving Section 4.2 (Statistical analysis of the transport properties of the porous medium) to Supplements, as it contains results that are tangential to the main message of the manuscript (see Supplementary

- 4 in the updated manuscript), and (b) the discussion of the absolute Reactive entropy removed altogether, now focusing on discussing the useful Reaction entropy, as this quantity can be measured experimentally.
- Section rewritten to allow for a clearer and more concise analysis of the obtained results, as well as their interpretation in terms of the correlation between transport self-organization in the field and the accompanying dissipative dynamics.
- A concluding Section added that frames our findings in the broader implications for real-world geochemical systems on an example of CO2 sequestration by mineralization, as a highly relevant example of an interaction between transport and dissolution-precipitation reaction (see Section 4.3, lines 651-678 in the updated manuscript). The proposed Section reads:

**4.3 Implications for subsurface carbon sequestration by mineralization**

The relevance of our study can be exemplified on a case of subsurface carbon sequestration by mineralization that involves injection of either dissolved or supercritical  $CO_2$  into reactive rocks, particularly mafic, ultramafic or carbonate, leading to CO2 mineralization and fixing carbon in the subsurface with little risk of escape to the atmosphere (Snaebjornsdottir et al., 2020, Ruprecht and Falta, 2012). For the supercritical CO2 injection, CO2 solubility in brine increases with pressure as the solute advances deeper into subsurface, thus leading to acidification of the brine layer in contact with the supercritical  $CO_2$  (Pradhan et al., 2025). This enriched layer, having a typical pH of 3-5, is denser than the surrounding resident brine, thus creating a negative buoyancy. This facilitates the transport of acidic brine downwards in the porous matrix (Ahmad et al., 2016). This acidic solution promotes the dissolution of silicate minerals, facilitating the subsequent  $CO_2$  mineralization by (a) consuming the hydrogen ions, which neutralizes the acidic solution and facilitates precipitation of carbonate minerals, and (b) providing cations that react with the dissolved  $CO_2$  to form stable carbonate minerals (Snaebiornsdottir et al., 2020). Thus, a complex reaction-transport interaction ensues following  $CO_2$  injection into subsurface, where the rate of  $CO_2$ mineralization is strongly affected by the preferential flow paths, either existing in the subsurface or emerged/altered due to dissolution of silicate minerals. Although  $CO_2$  mineralization in mafic rocks involves dissolved cations different from  $Ca^{2+}$  as in our study ( $Mq^{2+}$ ,  $Fe^{2+}$ , etc.), nevertheless the mechanism follows the same traits, thus allowing us to speculate on this subject based on our study.

The existence of dominant preferential flow paths and their intensification due to reaction-transport interaction can facilitate  $CO_2$  injection and may be favorable for carbon sequestration as it allows decreasing the hydraulic power per unit flow rate required for  $CO_2$  injection (Oldenburg, 2001). This is confirmed by Figure 6c in the updated manuscript that shows a decrease in the normalized Percolative entropy with the intensification of preferential flow paths due to either an increase in the field heterogeneity  $\sigma_0^2$  or the computational time  $\tilde{t}$ , which signifies a decrease in the free-energy dissipation due to flow percolation. On the other hand, for  $CO_2$  mineralization, the existence of dominant preferential flow paths may inhibit the overall extent of reaction per unit flow rate due to channelization of reactive species that limits their exposure to calcite (Harrison et al., 2016). This is confirmed by Figure 6a that shows a decrease in the normalized Reactive entropy with the intensification of preferential flow paths, signifying a decrease in the overall reaction rate. To summarize, for a given pressure head, an injection into a more heterogeneous matrix will result in a higher injection rate, while a more homogeneous domain will yield a higher mineralization rate, thus exemplifying the resulting trade-off in the injection strategy.

**310 (f) **Conclusions**:**

Rewritten to better convey the key message of the manuscript regarding the correlation between the transport selforganization and the accompanying dissipative dynamics during chemical weathering, as well as to emphasize the geophysical aspects of chemical weathering and the implications of our study for real-world geophysical scenario of subsurface CO2 sequestration by mineralization.

**Appendix A: Employing Shannon entropy to quantify transport self-organization**

To revise the basic concept of information entropy, let us consider a single probabilistic event, having a number of possible outcomes that are assumed equally probable. For example, in the case of a single symbol transmitted in the Morse code, we have two possible outcomes (realizations) for this event - a dash and a dot (omitting the intermission between the transmitted letters). Employing the indices 0 and 1 to denote parameters before and after the event realization, initially, we have  $N_0 = 2$  equally possible outcomes of the message transmission event and zero initial information  $I_0 = 0$ . Following the transmission of a single symbol, we have a single realization of the event - a dash or a dot, thus  $N_1 = 1$ , and a non zero information due to the receipt of a symbol  $I_1 \neq 0$ .

In case of a sequence of events enumerated 1...n, having a respective number of outcomes  $N_{0_1}, N_{0_2}, ..., N_{0_n}$  (consider a word that consists of n symbols in the Morse alphabet), the total number of possible outcomes is  $N = N_{0_1} \cdot N_{0_2} \cdot ... \cdot N_{0_n}$ . Intuitively, we would expect the information measure to be an additive parameter so that the amount of information contained in a word transmitted with the help of the Morse code would be equal to the sum of the amounts of information for each symbol that constitutes the word, that is  $I(N) = I(N_{0_1} \cdot N_{0_2} \cdot ... \cdot N_{0_n}) = I(N_{0_1}) + I(N_{0_2}) + ... + I(N_{0_n})$ . This can be fulfilled by choosing  $I = b \ln N$ , where b is an arbitrary constant parameter that amounts to a choice of a unit of measure (in fact, Shannon (1948) gave a formal proof that the logarithm function is the only possible relation between I and I0 that possesses, in addition to additivity, also continuity and monotonicity properties). For a binary system that consists of only two symbols, such as the dash and the dot in the Morse alphabet, a transmitted word of a total I1 symbols has I2 I3 I4 I5 I5 I6 possible realizations. Taking a single transmitted symbol as one unit of information, we demand that I5 I7 I8 to obtain I8 be and I8 be called bits (an abbreviation from a binary system is defined as a bit, the corresponding units of information I2 will be called bits (an abbreviation from binary digit).

Now, assume that each of the final amount of different symbols that constitute a message has its own relative occurrence frequency (the respective probability of finding a specific symbol at a specific place in a sequence), such as the case of different letters in an alphabet. In the case of a Morse code, assume that the transmitted n-symbol word consists of  $n_1$  dashes and  $n_2$  dots, so that  $n_1 + n_2 = n$ . Using some results from combinatorics, it can be shown that

$$S = I/n = -\Sigma_i \, p_i \log_2 p_i \tag{1}$$

where S is the information entropy (also referred to as the Shannon entropy) per symbol and  $p_i = n_i/n$ , i = 1,2 are the relative occurrence frequencies of both symbols. Maximum Shannon entropy obtained during the transmission of a message is when the relative occurrence frequency of each symbol is identical p = 1/2. It is easily shown that this maximum value equals to  $S_{max} = -\log_2 p = 1$ . The relation (1) can be easily generalized for any number of possible outcomes per event. Thus, a definition of the information entropy is obtained that possesses an additivity property similar to physical properties such as entropy, energy, and mass. An example that may be seen as a consequence of the results obtained with the help of the information theory is that in modern communication methods, the more common alphabet letters are encoded in such a way that their information content is minimized in order to facilitate the transmission process. Thus, in a Morse code, a letter E is encoded by a single dot, while J is a dot followed by three dashes. For more details on the topic of Shannon entropy, see Shannon (1948).

This definition resembles the definition physical entropy in statistical mechanics, as defined by Gibbs, where the logarithm in Eq. (1) is to the base of e and the sum is multiplied by the Boltzmann constant. The statistical definition of physical entropy characterizes the number of possible microstates of a system that are consistent with its macroscopic thermodynamic properties, which constitute the macrostate of the system. Thus, in the case of a gas consisting of a large number of molecules in a container, a microstate of the system consists of the position and momentum of each molecule as it moves within the container, colliding with other molecules and container walls. A multitude of such microstates correspond to a single macroscopic state of the system, defined by its pressure and temperature. The parameter  $p_i$  in this case corresponds to a probability that a microstate i occurs during the system's fluctuations. According to the second law of Thermodynamics, the entropy of an isolated system reaches its maximum value at equilibrium, where gradients of thermodynamic parameters are depleted by dissipative forces and the measure of order in the system is at its minimum. In this case, each microstate is equally likely and  $p_i$  is simply the inverse of the total number of microstates (Kondepudi and Prigogine, 1998). Returning to the context of communication theory, maximum entropy is obtained during the transmission of a message of a length n when an alphabet with the largest number of symbols is employed, and the relative occurrence frequency of each symbol is identical.

To characterize the emergence and evolution of transport self-organization in a computational field as the dissolutionprecipitation reactive processes in the field advance, we adopt a straightforward use of the Shannon entropy, in a similar vein to Zehe et al (2021) and Shavelzon and Edery (2024), where it was employed in the context of characterizing self-organization in non-reactive flow in a heterogeneous porous medium. We calculate the Shannon entropy of particle distribution in the direction transverse to flow at a dimensionless time  $\tilde{t}$  as a function of distance from the inlet x,  $S(x,\tilde{t})$ , using the same formula as in (1), reprinted here as

$$S(x, \tilde{t}) = -\sum_{i} p_{i} \log_{2} p_{i} \tag{2}$$

where  $p_i = N_i/N_{tot}$ ,  $i = 1...N_y$  is the relative concentration distribution at  $x, \tilde{t}$  in the direction transverse to flow (here,  $N_i$  is the current number of particles of a specific specie in the i-th cell in the vertical direction at x and  $N_{tot}$  is the total number of particles of that specie in all cells at x, taken at a dimensionless time  $\tilde{t}$ ). This quantity has an upper bound of  $S_{max} = \log_2 N_y$ , corresponding to the state of maximum entropy, or homogeneity of transport in the field, where the same number of particles has visited each of the  $N_y$  computational cells at a given distance from inlet x; the obvious lower bound of 0 corresponds to the state of maximum transport spatial organization in the field, namely all particles follow a single preferential flow path (Zehe et al (2021) and Shavelzon and Edery (2024)).

**385 References:**

- Shavelzon and Edery (2024), Shannon entropy of transport self-organization due to dissolution-precipitation reaction at varying Peclet numbers in initially homogeneous porous media,
- Shannon (1948), A Mathematical Theory Of Communication.
- Zehe et al (2021), Preferential Pathways for Fluid and Solutes in Heterogeneous Groundwater Systems.
- Edery and Stolar (2021), Feedback mechanisms between precipitation and dissolution reactions across randomly heterogeneous conductivity fields.
- Edery et al (2011), Dissolution and precipitation dynamics during dedolomitization.
- Singurindy and Berkowitz (2003), Flow, dissolution, and precipitation in dolomite.
- Al-Khulaifi et al(2017), Reaction Rates in Chemically Heterogeneous Rock: Coupled Impact of Structure and Flow Properties Studied by X-ray Microtomography
- Manahan (2000), Environmental Chemistry (9th edn).
- Amorsson (1981), Mineral Deposition From Icelandic Geothermal Waters: Environmental and Utilization Problems.
- Snaebjornsdottir et al (2020), Carbon dioxide storage through mineral carbonation.
- Ruprecht and Falta (2012), COMPARISON OF SUPERCRITICAL AND DISSOLVED CO2 INJECTION SCHEMES.
- Pradhan et al (2025), Determination of CO2 solubility in brines and produced waters of various salinities for CO2
  EOR and storage applications.

- Ahmad et al (2016), Injection of CO2-saturated brine in geological reservoir: A way to enhanced storage safety.
- Oldenburg et al (2001), Carbon Sequestration with Enhanced Gas Recovery: Identifying Candidate Sites for Pilot Study.
- Cao et al (2013), Dynamic alterations in wellbore cement integrity due to geochemical reactions in CO2-rich environments.
- Haugen et al (2023), Multi-scale dissolution dynamics for carbon sequestration in carbonate rock samples.
- Harrison et al (2015), The impact of evolving mineral-water-gas interfacial areas on mineral-fluid reaction rates in unsaturated porous media
- Kondepudi and Prigogine (1998), Modern Thermodynamics: From Heat Engines to Dissipative Structures.
- Kaufmann et al (2016), Dissolution and precipitation of fractures in soluble rock

- Szawello et al (2024), Quantifying dissolution dynamics in porous media using a spatial flow focusing profile.
- Liu et al (2005), Dissolution-induced preferential flow in a limestone fracture.
- Jones and Detwiller (2016), Fracture sealing by mineral precipitation: The role of small-scale mineral heterogeneity.
  - Kohlhaas et al (2025), Numerical Investigation of Preferential Flow Paths in Enzymatically Induced Calcite Precipitation supported by Bayesian Model Analysis.
  - Deutsch (1997), GSLIB Geostatistical Software Library and User's Guide.
  - Morse and Mackenzie (1990), Geochemistry of Sedimentary Carbonates
    - Kreft and Zuber (1978), On the physical meaning of the dispersion equation and its solutions for different initial and boundary conditions.
    - Perez et al (2019), Reactive Random Walk Particle Tracking and Its Equivalence With the Advection-Diffusion-Reaction Equation.
  - Risken (1996), The Fokker-Planck Equation: Methods of Solution and Applications.
    - Press et al (1992), Numerical Recipes in C: The Art of Scientific Computing.
    - Kloeden (1992), Numerical solution of stochastic differential equations.
    - Domenico and Schwartz, Physical and Chemical Hydrogeology, 2nd Edition.
    - Le Borgne et al (2008), Lagrangian Statistical Model for Transport in Highly Heterogeneous Velocity Fields.
- Gudagninin and Neumann (1999), Nonlocal and localized analyses of conditional mean steady state flow in bounded,
  randomly nonuniform domains: 1. Theory and computational approach.

**Linking chemical weathering, evolution of preferential flow paths and transport self-organization in porous media using non-equilibrium thermodynamics**

Evgeny Shavelzon1, Erwin Zehe2, and Yaniv Edery1

**Correspondence:** Yaniv Edery(yanivedery@technion.ac.il)

**Replies to Reviewer 2**

**Reviewer 2**

The manuscript by Shavelzon et al. aims to establish a novel connection between chemical weathering, transport self-organization, and non-equilibrium thermodynamics by using a particle-tracking reactive transport model in 2D porous media with varying heterogeneity. This topic would likely be of interest to researchers seeking better frameworks to describe the formation of preferential pathways in the subsurface driven by water-rock interactions. However, the current version of the manuscript is difficult to read and interpret, due to excessive technical density, insufficient narrative structure, and limited discussion of the model assumptions and broader scientific implications. I think major revisions are needed before the manuscript can be considered for publication.

We thank the reviewer for recognizing the potential relevance of our work to the HESS readership. The comments from the reviewer are well received and were extremely helpful in clarifying and improving our study. We are also thankful to the reviewer for indicating the need to improve the readability of the manuscript and its relevance for the geophysical community by clarifying it and making it more concise, as well as by discussing the broader implications of our research for real-world geochemical systems. We fully acknowledge this need and are committed to amending the manuscript accordingly.

1. Excessive length, technical complexity and abstract terminology. The paper dives into lengthy derivations and modeling details early on (especially Sections 2-3), but without sufficient explanations for why this framework matters or what the reader should expect to learn. This makes it difficult to parse how the pieces fit together—especially for readers unfamiliar with entropy generation in porous media. A conceptual figure or roadmap either at the end of the Introduction or at the beginning of the Methods would help guide the readers.

<sup>1Faculty of Civil and Environmental Engineering, Technion - Israeli Institute of Technology, Haifa, Israel

<sup>2Karlsruhe Institute of Technology (KIT), Institute of Water and River Basin Management, Karlsruhe, Germany

We sincerely thank the reviewer for pointing out the need to improve the readability of the manuscript, clarify it and make it more concise. We fully acknowledge this need and are committed to amending the manuscript accordingly. Please find attached the summary of our suggestions regarding shortening the manuscript, as well as improving its overall clarity and readability, organized by chapter. All of these suggestions have been implemented in the updated manuscript draft.

**(a) Abstract:**

Rewritten to better convey the key message of the manuscript regarding the correlation between the transport self-organization and the accompanying dissipative dynamics during chemical weathering, as well as to emphasize the geophysical aspects of chemical weathering and the implications of our study for real-world geophysical scenario of subsurface CO2 sequestration by mineralization (see lines 1–21 in the updated manuscript):

**Abstract.** Chemical weathering of soil and rock is a complex geophysical process during which the reaction and transport processes in the porous medium interact, causing erosion of the medium. This process is ubiquitous in geophysical systems and can be encountered, among others, in formation of karst systems, subsurface carbon seauestration and surface weathering of river beds. A common outcome of chemical weathering is the emergence and intensification of preferential flow paths, where the weathering alters the transport properties of the rock, thus creating coupling between transport and reaction. While numerous approaches have been undertaken to simulate this complex interaction, still a need exists for a unified framework able to correlate the emergence of preferential flow paths due to reaction-transport interaction with the associated dissipative dynamics. Here we propose such a framework considering the case of subsurface chemical weathering of calcite porous rock undergoing reversible dissolution-precipitation reaction, and apply non-equilibrium thermodynamics to analyze the ensuing reactiontransport interaction in this geophysical scenario. We identify the entropy generation sources, attributed to the dissipative processes inherent to this physical scenario and show a clear correlation between the emergence and intensification of preferential flow paths and the accompanying dissipative dynamics, where the evolution of the emerging paths leads to a decrease in the free-energy dissipation rate due to flow percolation, mixing of chemical constituents and reaction. This indicates that the emergence of preferential flow paths due to chemical weathering in geophysical systems represents an energetically-preferred state of the system that can be considered a manifestation of the minimum energy dissipation principle. Our analysis implies that, for a given pressure head, a more homogeneous porous matrix will result in less pronounced preferential flow paths, along with lower flow and higher mineralization rates. On the other hand, for a highly heterogeneous matrix dominant preferential flow paths will be obtained, along with higher flow and lower mineralization rates, Considering these aspects for carbon sequestration where acidified brine leads to carbon mineralization, we conclude that, for a given pressure head, an injection into a more heterogeneous matrix will result in a higher injection rate, while a more homogeneous domain will yield a higher mineralization rate, thus exemplifying the resulting trade-off in the injection strategy.

**(b) Chapter 1 (Introduction):**

Rewritten to improve clarity of the exposition: beginning from Viewing chemical weathering as a non-equilibrium thermodynamic process, we move to Overall review of analysis of non-equilibrium systems and then to Self-organization as a manifestation of non-equilibrium state. We proceed with Application of non-equilibrium thermodynamics to transport in porous media and conclude with Objectives of our study. See updated manuscript.

**(c) Chapter 2 (Methodology):**

A conceptual flowchart added at the beginning of Chapter 2 (Methodology) - see Figure 1 in the current document,
 as well as Figure 1 in the *updated* manuscript. This flowchart details the various stages of the employed methodology.

**Figure 1.** Roadmap chart for the research procedure.

ogy, from creating a realization of hydraulic conductivity field to numerical simulation to calculating the Shannon entropy and the physical entropy sources employing data obtained from simulation. The flowchart also presents graphically the research question of the manuscript, that concerns the correlation between the emergence of preferential flow paths (transport self-organization quantified using Shannon entropy) and the free-energy dissipation due to flow percolation, mixing and reaction (physical entropy sources). Each graphical block (shown by different colors in the flowchart) references the specific section where it is explained in details (section numbers have been adjusted to reflect the suggested manuscript amendments in the updated manuscript). See lines 137–142 in the updated manuscript.

- Section 2.2 (Chemical reaction model) rewritten and expanded to allow for a clearer description of the chemical model and its underlying assumptions. Also, we suggest minor corrections/additions to Chapter 2 as a whole to improve the overall presentation of methodology. See our suggestion for the updated Chapter 2 in the updated manuscript, specifically Section 2.2 (lines 193-221).
- A short section on the Lagrangian particle tracking model validation in 2D added (in Shavelzon and Edery, 2024 the validation was done for the 1D scenario) - see Supplementary 1 in the updated manuscript.
- Abstract terminology throughout the manuscript simplified where possible such as the Langevin equation paraphernalia (lines 77–85 in the updated manuscript) as related to purely technical aspects of our study and unnecessary for conveying the key message of the manuscript.

**85 (d) Chapter 3 (Non-equilibrium thermodynamic framework):**

- Section 3.2 (Calculation of entropy generation terms) moved to Supplements, as it basically contains technical details that are not critical for understanding of the manuscript (See Supplementary 3 in the updated manuscript).
- A short chapter that explains in simple terms the topic of dissipative processes in non-equilibrium thermodynamics added to the Supplementary. See Supplementary 2 in the updated manuscript, also Section 3.1 there (lines 282–311).
- A short section added that briefly explains the concept of the Shannon entropy and its use for quantifying self-organization in our physical scenario along the lines of Section 3.2 and Supplementary 2 in Shavelzon and Edery (2024) (currently, the manuscript draft references this publication, where this subject is treated extensively). See Section 3.2 in the updated manuscript (lines 371—401). Also, notice that references to Supplementary 2 in Shavelzon and Edery (2024), which contains a brief background on the Shannon entropy, are given in the appropriate places in the manuscript. In that way, both the Shannon and the physical entropy, as well as their relevance in our manuscript, are explained before any use of them is implemented in the Results. Full text of Supplementary 2 in Shavelzon and Edery (2024) is given in the Appendix A: Employing Shannon entropy to quantify transport self-organization in the current document.
- Figure that shows graphically the sources of inflow and dissipation of free energy in subsurface reactive flow replaced with an improved version that explains more clearly the sources of free-energy inlfow into the system

Figure 2. Sources of inflow and dissipation of free energy in subsurface reactive flow.

and its dissipation (see Figure 2 in the current document, as well as Figure 2 in the *updated* manuscript). This figure visualizes the main result of Section 3.1: entropy generation, which is directly related to dissipation of the free-energy of the system, occurs in dissipative processes where the gradient of a thermodynamic potential drives the thermodynamic flux. Thus, the gradient of hydraulic head drives the mass flux, the gradient of the chemical potential drives the molar fluxes of the chemical species and the partial molar Gibbs energy drives the extent of reaction. See Section 3.1 (lines 283–310) and Supplementary 2 in the updated manuscript for detailed explanation.

Abstract terminology throughout the manuscript simplified where possible - such as differential operators used in the exposition of non-equilibrium thermodynamic framework (lines 345–360 in the updated manuscript) - as related to purely technical aspects of our study and unnecessary for conveying the key message of the manuscript.

**(e) Section 4 (Results):**

Shortened by (a) moving Section 4.2 (Statistical analysis of the transport properties of the porous medium) to
 Supplements, as it contains results that are tangential to the main message of the manuscript (see Supplementary
in the updated manuscript), and (b) the discussion of the absolute Reactive entropy removed altogether, now focusing on discussing the useful Reaction entropy, as this quantity can be measured experimentally.

- Section rewritten to allow for a clearer and more concise analysis of the obtained results, as well as their interpretation in terms of the correlation between transport self-organization in the field and the accompanying dissipative dynamics.
- A concluding Section added that frames our findings in the broader implications for real-world geochemical systems on an example of CO2 sequestration by mineralization, as a highly relevant example of an interaction between transport and dissolution-precipitation reaction (see Section 4.3, lines 651-678 in the updated manuscript). The proposed Section reads:

**4.3 Implications for subsurface carbon sequestration by mineralization**

The relevance of our study can be exemplified on a case of subsurface carbon sequestration by mineralization that involves injection of either dissolved or supercritical CO2 into reactive rocks, particularly mafic, ultramafic or carbonate, leading to CO2 mineralization and fixing carbon in the subsurface with little risk of escape to the atmosphere (Snaebjornsdottir et al., 2020, Ruprecht and Falta, 2012). For the supercritical CO2 injection, CO2 solubility in brine increases with pressure as the solute advances deeper into subsurface, thus leading to acidification of the brine layer in contact with the supercritical  $CO_2$  (Pradhan et al., 2025). This enriched layer, having a typical pH of 3-5, is denser than the surrounding resident brine, thus creating a negative buoyancy. This facilitates the transport of acidic brine downwards in the porous matrix (Ahmad et al., 2016). This acidic solution promotes the dissolution of silicate minerals, facilitating the subsequent  $CO_2$  mineralization by (a) consuming the hydrogen ions, which neutralizes the acidic solution and facilitates precipitation of carbonate minerals, and (b) providing cations that react with the dissolved  $CO_2$  to form stable carbonate minerals (Snaebjornsdottir et al., 2020). Thus, a complex reaction-transport interaction ensues following CO2 injection into subsurface, where the rate of CO2 mineralization is strongly affected by the preferential flow paths, either existing in the subsurface or emerged/altered due to dissolution of silicate minerals. Although  $CO_2$  mineralization in mafic rocks involves dissolved cations different from  $Ca^{2+}$  as in our study ( $Ma^{2+}$ ,  $Fe^{2+}$ , etc.), nevertheless the mechanism follows the same traits, thus allowing us to speculate on this subject based on our study.

The existence of dominant preferential flow paths and their intensification due to reaction-transport interaction can facilitate  $CO_2$  injection and may be favorable for carbon sequestration as it allows decreasing the hydraulic power per unit flow rate required for  $CO_2$  injection (Oldenburg, 2001). This is confirmed by Figure 6c in the updated manuscript that shows a decrease in the normalized Percolative entropy with the intensification of preferential flow paths due to either an increase in the field heterogeneity  $\sigma_0^2$  or the computational time  $\tilde{t}$ , which signifies a decrease in the free-energy dissipation due to flow percolation. On the other hand, for  $CO_2$  mineralization, the existence of dominant preferential flow paths may inhibit the overall extent of reaction per unit flow rate due to channelization of reactive species that limits their exposure to calcite (Harrison et al., 2016). This is confirmed by Figure 6a that shows a decrease in the normalized Reactive entropy with the intensification of preferential flow paths, signifying a decrease in the overall reaction rate. To summarize, for a given pressure head, an injection into a more heterogeneous matrix will result in a higher injection rate, while a more homogeneous domain will yield a higher mineralization rate, thus exemplifying the resulting trade-off in the injection strategy.

**(f) **Conclusions**:**

Rewritten to better convey the key message of the manuscript regarding the correlation between the transport selforganization and the accompanying dissipative dynamics during chemical weathering, as well as to emphasize the geophysical aspects of chemical weathering and the implications of our study for real-world geophysical scenario of subsurface CO2 sequestration by mineralization.

2. Unclear assumptions in reactive transport setup and their implications for the conclusions. The authors employ a simplified reactive system consisting of only two solute species (H+ and H2CO3) and a single dissolution–precipitation reaction involving calcite. Important geochemical complexities, such as full aqueous speciation (e.g., bicarbonate, carbonate), are omitted. I think it is necessary to clarify the rationale for these assumptions and discuss their implications for the applicability of results to real-world weathering environments.

We are sincerely thankful to the reviewer for indicating the need to provide a clear explanation for the origins of our chemical reaction model. This simplified chemical representation follows previous work by Edery et al. (2011), where this formulation is treated extensively, and was later employed in Edery et al. (2021) and Shavelzon and Edery (2024). The purpose of this simplified chemical reaction setup is to capture the qualitative dynamics of the complex process of geochemical weathering, thus making it possible to simulate and analyze the complex reversible behavior using a relatively unsophisticated model. The somewhat brief description of the model given in the manuscript follows from the desire to shorten the manuscript length, therefore a reference to Shavelzon and Edery (2024) was given in the beginning of Section 2. To allow for a clearer description of the chemical model, we propose dedicating Section 2.2 solely to the chemical reaction model, rewriting and expanding it, and putting the description of the kinetic aspects and reaction-transport interaction implementation in Section 2.3. Also, we suggest minor corrections/additions to Chapter 2 as a whole to improve the overall presentation of methodology. See our suggestion for the updated Chapter 2 in the updated manuscript (specifically, see Section 2.2 in lines 193-221).

3. Conclusions are too technically narrow. It would better for the authors to frame their findings in the broader implications for real-world geochemical systems, such as rock weathering in karst systems or channelization in CO2-enhanced weathering. Discussing such applications would help connect the modeling results to field-scale processes and enhance the overall significance of the study.

We again thank the reviewer for indicating the necessity to broaden the implications of our study to real-world geochemical processes and propose discussing the implications of the study on an example of subsurface CO2 sequestration by mineralization, as a highly relevant example of an interaction between transport and dissolution-precipitation reaction. A concluding Section was added to Results that frames our findings in the broader implications for real-world geochemical systems on an example of CO2 sequestration by mineralization, as a highly relevant example of an interaction between transport and dissolution-precipitation reaction (see our repl to the previous item in the current document, as well as Section 4.3, lines 651-678 in the updated manuscript).

**4. Minor comments:**

- Abstract, Line 2: "that" -> "which"; add "the" before "emergence".
- Introduction, Line 17: It is not appropriate to refer to chemical weathering as a geophysical phenomenon. Consider revising to "geochemical process".
- Line 663: typo "expend Figure 8biture for the network"

We again thank the reviewer for indicating these typos. These will be fixed in the updated draft.

**Appendix A: Employing Shannon entropy to quantify transport self-organization**

To revise the basic concept of information entropy, let us consider a single probabilistic event, having a number of possible outcomes that are assumed equally probable. For example, in the case of a single symbol transmitted in the Morse code, we have two possible outcomes (realizations) for this event - a dash and a dot (omitting the intermission between the transmitted letters). Employing the indices 0 and 1 to denote parameters before and after the event realization, initially, we have  $N_0 = 2$  equally possible outcomes of the message transmission event and zero initial information  $I_0 = 0$ . Following the transmission of a single symbol, we have a single realization of the event - a dash or a dot, thus  $N_1 = 1$ , and a non zero information due to the receipt of a symbol  $I_1 \neq 0$ .

In case of a sequence of events enumerated 1...n, having a respective number of outcomes  $N_{0_1}, N_{0_2}, ..., N_{0_n}$  (consider a word that consists of n symbols in the Morse alphabet), the total number of possible outcomes is  $N = N_{0_1} \cdot N_{0_2} \cdot ... \cdot N_{0_n}$ . Intuitively, we would expect the information measure to be an additive parameter so that the amount of information contained in a word transmitted with the help of the Morse code would be equal to the sum of the amounts of information for each symbol that constitutes the word, that is  $I(N) = I(N_{0_1} \cdot N_{0_2} \cdot ... \cdot N_{0_n}) = I(N_{0_1}) + I(N_{0_2}) + ... + I(N_{0_n})$ . This can be fulfilled by choosing  $I = b \ln N$ , where b is an arbitrary constant parameter that amounts to a choice of a unit of measure (in fact, Shannon (1948) gave a formal proof that the logarithm function is the only possible relation between I and I0 that possesses, in addition to additivity, also continuity and monotonicity properties). For a binary system that consists of only two symbols, such as the dash and the dot in the Morse alphabet, a transmitted word of a total I1 symbols has I2 symbols has I3 possible realizations. Taking a single transmitted symbol as one unit of information, we demand that  $I = b \ln N = n$  to obtain  $b = \log_2 e$

and  $I = \log_2 N$ . Since a single position in a sequence of symbols in a binary system is defined as a bit, the corresponding units of information I will be called bits (an abbreviation from  $binary\ digit$ ).

Now, assume that each of the final amount of different symbols that constitute a message has its own relative occurrence frequency (the respective probability of finding a specific symbol at a specific place in a sequence), such as the case of different letters in an alphabet. In the case of a Morse code, assume that the transmitted n-symbol word consists of  $n_1$  dashes and  $n_2$  dots, so that  $n_1 + n_2 = n$ . Using some results from combinatorics, it can be shown that

$$S = I/n = -\Sigma_i \ p_i \log_2 p_i \tag{1}$$

where S is the information entropy (also referred to as the Shannon entropy) per symbol and  $p_i = n_i/n$ , i = 1,2 are the relative occurrence frequencies of both symbols. Maximum Shannon entropy obtained during the transmission of a message is when the relative occurrence frequency of each symbol is identical p = 1/2. It is easily shown that this maximum value equals to  $S_{max} = -\log_2 p = 1$ . The relation (1) can be easily generalized for any number of possible outcomes per event. Thus, a definition of the information entropy is obtained that possesses an additivity property similar to physical properties such as entropy, energy, and mass. An example that may be seen as a consequence of the results obtained with the help of the information theory is that in modern communication methods, the more common alphabet letters are encoded in such a way that their information content is minimized in order to facilitate the transmission process. Thus, in a Morse code, a letter E is encoded by a single dot, while J is a dot followed by three dashes. For more details on the topic of Shannon entropy, see Shannon (1948).

This definition resembles the definition physical entropy in statistical mechanics, as defined by Gibbs, where the logarithm in Eq. (1) is to the base of e and the sum is multiplied by the Boltzmann constant. The statistical definition of physical entropy characterizes the number of possible microstates of a system that are consistent with its macroscopic thermodynamic properties, which constitute the macrostate of the system. Thus, in the case of a gas consisting of a large number of molecules in a container, a microstate of the system consists of the position and momentum of each molecule as it moves within the container, colliding with other molecules and container walls. A multitude of such microstates correspond to a single macroscopic state of the system, defined by its pressure and temperature. The parameter  $p_i$  in this case corresponds to a probability that a microstate i occurs during the system's fluctuations. According to the second law of Thermodynamics, the entropy of an isolated system reaches its maximum value at equilibrium, where gradients of thermodynamic parameters are depleted by dissipative forces and the measure of order in the system is at its minimum. In this case, each microstate is equally likely and  $p_i$  is simply the inverse of the total number of microstates (Kondepudi and Prigogine, 1998). Returning to the context of communication theory, maximum entropy is obtained during the transmission of a message of a length n when an alphabet with the largest number of symbols is employed, and the relative occurrence frequency of each symbol is identical.

To characterize the emergence and evolution of transport self-organization in a computational field as the dissolution-precipitation reactive processes in the field advance, we adopt a straightforward use of the Shannon entropy, in a similar vein to Zehe et al (2021) and Shavelzon and Edery (2024), where it was employed in the context of characterizing self-organization in non-reactive flow in a heterogeneous porous medium. We calculate the Shannon entropy of particle distribution in the direction transverse to flow at a dimensionless time  $\tilde{t}$  as a function of distance from the inlet x,  $S(x,\tilde{t})$ , using the same formula as in (1), reprinted here as

$$S(x,\tilde{t}) = -\sum_{i} p_{i} \log_{2} p_{i} \tag{2}$$

where  $p_i = N_i/N_{tot}$ ,  $i = 1...N_y$  is the relative concentration distribution at  $x, \tilde{t}$  in the direction transverse to flow (here,  $N_i$  is the current number of particles of a specific specie in the i-th cell in the vertical direction at x and  $N_{tot}$  is the total number of particles of that specie in all cells at x, taken at a dimensionless time  $\tilde{t}$ ). This quantity has an upper bound of  $S_{max} = \log_2 N_y$ , corresponding to the state of maximum entropy, or homogeneity of transport in the field, where the same number of particles has visited each of the  $N_y$  computational cells at a given distance from inlet x; the obvious lower bound of 0 corresponds to the state of maximum transport spatial organization in the field, namely all particles follow a single preferential flow path (Zehe et al (2021) and Shavelzon and Edery (2024)).

**References:**

- Shavelzon and Edery (2024), Shannon entropy of transport self-organization due to dissolution-precipitation reaction at
  varying Peclet numbers in initially homogeneous porous media.
  - Shannon (1948), A Mathematical Theory Of Communication.
  - Zehe et al (2021), Preferential Pathways for Fluid and Solutes in Heterogeneous Groundwater Systems.
  - Edery and Stolar (2021), Feedback mechanisms between precipitation and dissolution reactions across randomly heterogeneous conductivity fields.
- Edery et al (2011), Dissolution and precipitation dynamics during dedolomitization.
  - Singurindy and Berkowitz (2003), Flow, dissolution, and precipitation in dolomite.
  - Al-Khulaifi et al(2017), Reaction Rates in Chemically Heterogeneous Rock: Coupled Impact of Structure and Flow Properties Studied by X-ray Microtomography
  - Manahan (2000), Environmental Chemistry (9th edn).
- Amorsson (1981), Mineral Deposition From Icelandic Geothermal Waters: Environmental and Utilization Problems.
  - Snaebjornsdottir et al (2020), Carbon dioxide storage through mineral carbonation.
  - Ruprecht and Falta (2012), COMPARISON OF SUPERCRITICAL AND DISSOLVED CO2 INJECTION SCHEMES.

- Pradhan et al (2025), Determination of CO2 solubility in brines and produced waters of various salinities for CO2 EOR and storage applications.
- Ahmad et al (2016), Injection of CO2-saturated brine in geological reservoir: A way to enhanced storage safety.
  - Oldenburg et al (2001), Carbon Sequestration with Enhanced Gas Recovery: Identifying Candidate Sites for Pilot Study.
  - Cao et al (2013), Dynamic alterations in wellbore cement integrity due to geochemical reactions in CO2-rich environments.
  - Haugen et al (2023), Multi-scale dissolution dynamics for carbon sequestration in carbonate rock samples.
- Harrison et al (2015), The impact of evolving mineral-water-gas interfacial areas on mineral-fluid reaction rates in unsaturated porous media.
  - Deutsch (1997), GSLIB Geostatistical Software Library and User's Guide.
  - Morse and Mackenzie (1990), Geochemistry of Sedimentary Carbonates
- Kreft and Zuber (1978), On the physical meaning of the dispersion equation and its solutions for different initial and
  boundary conditions.
  - Perez et al (2019), Reactive Random Walk Particle Tracking and Its Equivalence With the Advection-Diffusion-Reaction
    Equation.
  - Risken (1996), The Fokker-Planck Equation: Methods of Solution and Applications.
  - Press et al (1992), Numerical Recipes in C: The Art of Scientific Computing.
- Kloeden (1992), Numerical solution of stochastic differential equations.
  - Domenico and Schwartz, Physical and Chemical Hydrogeology, 2nd Edition.
  - Le Borgne et al (2008), Lagrangian Statistical Model for Transport in Highly Heterogeneous Velocity Fields.
  - Gudagninin and Neumann (1999), Nonlocal and localized analyses of conditional mean steady state flow in bounded, randomly nonuniform domains: 1. Theory and computational approach.

---

## Author Response (AR2)

| Following the decision by the respectable Editor to publish the manuscript as is after the Major |
|--------------------------------------------------------------------------------------------------|
| revision round, please find attached the required files for publication.                         |

Sincerely,

Evgeny Shavelzon, Erwin Zehe and Yaniv Edery.